# Dairy Wastewaters to Promote Mixotrophic Metabolism in *Limnospira* (*Spirulina*) *platensis*: Effect on Biomass Composition, Phycocyanin Content, and Fatty Acid Methyl Ester Profile

**DOI:** 10.3390/life15020184

**Published:** 2025-01-26

**Authors:** Luca Baraldi, Luca Usai, Serenella Torre, Giacomo Fais, Mattia Casula, Debora Dessi, Paola Nieri, Alessandro Concas, Giovanni Antonio Lutzu

**Affiliations:** 1Department of Life Sciences, University of Modena and Reggio Emilia, Via Giuseppe Campi 287, 41123 Modena, Italy; lucabara01@gmail.com; 2Teregroup Srl, Via David Livingstone 37, 41123 Modena, Italy; luca.usai@teregroup.net; 3Department of Pharmacy, University of Pisa, Via Bonanno Pisano 12, 56126 Pisa, Italy; serenella.torre@phd.unipi.it (S.T.); paola.nieri@unipi.it (P.N.); 4Department of Mechanical, Chemical and Materials Engineering, University of Cagliari, Piazza d’Armi, 09123 Cagliari, Italy; giacomo.fais@unica.it (G.F.); mattia.casula@unica.it (M.C.); alessandro.concas@unica.it (A.C.); 5Interdepartmental Center of Environmental Science and Engineering (CINSA), University of Cagliari, Via San Giorgio 12, 09124 Cagliari, Italy; 6Department of Life and Environmental Sciences, University of Cagliari, Cittadella Universitaria, Blocco A, SP8 Km 0.700, 09042 Monserrato, Italy; deboradessi95@gmail.com

**Keywords:** cheese whey, buttermilk, dairy wastewater, *Limnospira platensis*, phycocyanin, FAME profile

## Abstract

This study explores the mixotrophic cultivation of *Limnospira platensis* using dairy byproducts, specifically scotta whey (SW), buttermilk wastewater (BMW), and dairy wastewater (DWW), to promote biomass production and enhance the composition of bioactive compounds. By assessing various concentrations (1%, 2%, and 4% v v^−1^) of these byproducts in a modified growth medium, this study aims to evaluate their effect on *L. platensis* growth, phycocyanin (C-PC) content, and fatty acid methyl ester (FAME) profiles. The results show that the optimal biomass production was achieved with 2% scotta and dairy wastewater, reaching maximum concentrations of 3.30 g L^−1^ and 3.19 g L^−1^, respectively. Mixotrophic cultivation led to increased C-PC yields, especially in buttermilk and dairy wastewater treatments, highlighting the potential for producing valuable pigments. Additionally, the FAME profiles indicated minimal changes compared to the control, with oleic and γ-linolenic acids being dominant in mixotrophic conditions. These findings support the viability of utilizing dairy byproducts for sustainable *L. platensis* cultivation, contributing to a circular bioeconomy while producing bioactive compounds of nutritional and commercial interest.

## 1. Introduction

Microalgae are a valuable and abundant source of numerous biologically active compounds, such as proteins, lipids, carbohydrates, vitamins, pigments, enzymes, fatty acids (FAs), polyphenols, peptides, bioplastics, biofertilizers, and biofules. These diverse substances hold significant potential for use in a wide range of industries [1,2,3]. These photosynthetic organisms are widely known for their potent antioxidant, immune-enhancing, antiviral, and antimicrobial properties, attributed to their bioactive compounds [4,5]. Antioxidants are essential for maintaining human health, as they help inhibit or reduce the oxidation of sensitive molecules, shielding the body from the damaging effects of free radicals [6]. Key antioxidant compounds found in microalgae include polyphenols, carotenoids (such as ß-carotene, astaxanthin, fucoxanthin, and lutein), polyunsaturated fatty acids (PUFAs), polysaccharides, and phycobiliproteins [1,7]. The lipid composition of microalgae is highly varied, with substantial research emphasizing their long-chain polyunsaturated fatty acids (LC-PUFAs), particularly omega-3 FAs (ω3-PUFAs) like docosahexaenoic acid (DHA, C22:6 ω3) and eicosapentaenoic acid (EPA, C20:5 ω3), which are renowned for their health-promoting properties. The production of EPA and DHA varies across microalgal species, depending on the type and cultivation conditions [8].

For over ten years, large-scale cultivation of various microalgae and Cyanobacteria strains has been actively pursued, yielding a protein-rich biomass known for its valuable bioactive and functional compounds [9,10]. The global nutraceutical market, valued USD 200.2 billion in 2017, is projected to grow at a 6.8% Compound Annual Growth rate (CAGR), reaching USD 317.3 billion by 2024 [11]. Based on this growth rate, the market is estimated to have reached USD 278.2 billion in 2022 and USD 297.1 billion in 2023. By 2029, the nutraceutical market is forecast to expand further to approximately USD 440.9 billion. These projections align with existing growth trends in the nutraceutical sector, driven by rising consumer interest in health and wellness products [12]. In the European market, microalgae are crucial to these industries, where species like *Chlorella* sp., *Chlamidomonas* sp., *Dunaliella* sp. *Haematococcus* sp. (Chlorophyta), *Nannochloropsis* sp. (Eustigmatophyceae), and *Limnospira* (formerly *Spirulina*) (Cyanobacteria) are widely used as dietary supplements in the food sector due to their high levels of digestible protein, balanced amino acid profile, and abundant vitamins, polysaccharides, and PUFAs [13]. These microalgae species significantly boost the nutritional quality of food products and animal feed by providing essential nutrients and valuable bioactive extracts [14,15,16]. Recent research has highlighted the strong antioxidant properties of a photosynthetic prokaryotic organism such as *Limnospira* platensis (formerly *Spirulina platensis*) (Cyano-bacteria), demonstrated both in vivo [17] and in vitro [18,19], emphasizing its ability to reduce oxidative stress. Additionally, *Limnospira* extracts, particularly phycobiliproteins, have shown promising anticancer and anti-inflammatory properties [20]. Despite the vast industrial potential of microalgae, the high cultivation costs remain a barrier to widespread commercialization [21]. One promising approach is to integrate the production of high-value biomass with agro-industrial waste treatment, as microalgae can efficiently remove pollutants such as nitrogen (N), phosphorous (P), and organic carbon from wastewater (WW), helping to lower cultivation costs. Previous studies demonstrated how wastes and effluents from the food industry, including the brewery and dairy sectors, can be regarded as promising low-cost and high-efficiency promoters for microalgae growth [22,23,24]. In this frame, agricultural byproducts and fruit wastes (such as sugarcane bagasse, banana peel, watermelon rind, and rice bran) have gained recent interest as an attractive option for sustainable culture media development [25,26]. Mixotrophic cultivation, favored by the presence of organic sources found in agro-industrial wastes, enhance microalgae biomass production [27]. However, this method is prone to contamination, making closed, sterilized photobioreactors (PBRs) more suitable than open ponds [9]. Although more costly, PBRs improve both biomass yield and the quality of microalgal biomass under mixotrophic conditions compared to autotrophic growth [10].

The dairy industry produces significant byproducts like scotta (SW), buttermilk (BMW), and residual dairy wastewater (DWW), residues generated during the production of various dairy goods including ricotta, butter, and general dairy processing [28,29]. Rich in lactose, which acts as a key carbon source; proteins; and fats, these byproducts represent potential substrates for mixotrophic cultivation of microalgae [23,27]. Historically regarded as waste with environmental implications, these byproducts—especially DWW—pose challenges due to the large volumes generated, which often exceed four times that of processed milk [30]. Recent studies have shown that microalgae can be cultivated in mixotrophic cultures using dairy byproducts, providing essential nutrients and reducing the need for costly chemical supplements [31,32].

*L. platensis* was grown under mixotrophic conditions with varying concentrations of cheese whey [23]. At a concentration of 0.8% v v^−1^ of this effluent, this cyanobacterium demonstrated accelerated growth, indicating its potential for rapid biomass production. Under these conditions, significantly higher levels of phycocyanin were produced compared to photoautotrophic conditions (3.52 mg mL^−1^ vs. 2.55 mg mL^−1^). The fatty acid methyl ester (FAME) profile in mixotrophic conditions showed only minor changes in FAs compared to the control. Notably, mixotrophic cultivation led to an increase in the percentage of γ-linolenic fatty acid ω-6. *Auxenochlorella protothecoides* (formerly *Chlorella protothecoides*) (Chlorophyta) was grown in ricotta cheese whey (scotta) to assess the viability of this dairy byproduct as a cost-effective substrate [33]. The mixotrophic cultures yielded greater biomass compared to the autotrophic ones; however, the latter exhibited higher cellular concentrations of chlorophyll and carotenoids. Nonetheless, the stress strategy implemented promoted carotenogenesis, facilitating the accumulation of astaxanthin and lutein/zeaxanthin. These findings indicate that by employing an appropriate stress strategy, it is possible to effectively regulate carotenogenesis, leading to the production of substantial quantities of valuable high-value compounds. Buttermilk was utilized as a carbon source to investigate the growth of the polyextremophile red microalga *Galdieria sulphuraria* (Rhodophyta) under mixotrophic and heterotrophic conditions in laboratory-scale flasks and a 13 L photobioreactor [34]. Experiments conducted in flasks under mixotrophic conditions with varying dilutions of buttermilk indicated that a dilution ratio of 40% v v^−1^ was optimal for biomass production. When *G. sulphuraria* was cultivated at this optimal dilution in a 13 L photobioreactor, the highest biomass productivity of 0.55 g L^−1^ d^−1^ was achieved under mixotrophic conditions. This study overall highlights the potential of lactose-containing substrates, such as buttermilk, as effective growth media of microalgae while revalorizing an industrial effluent.

Considering the large-scale production of dairy byproducts and their potential environmental consequences, as well as the high market demand for phycobiliproteins in nutraceutical and pharmaceutical industries, this study seeks to investigate the production of phycobiliproteins and lipids from *L. platensis* cultivated under mixotrophic conditions using different concentrations of scotta, buttermilk, and residual DWW.

## 2. Materials and Methods

### 2.1. Inoculum and Culture Media Preparation

The *Limnospira platensis* strain SAG 21.99 used in this research was sourced from the culture collection of algae at the Gottingen University, Germany [35]. The cells were cultivated in a modified Jourdan Medium (JM), with the following composition per liter: 5 g NaHCO_3_; 1.6 g KOH, 5 g NaNO_3_; 0.027 g CaCl_2_·2H_2_O g; 0.4 g K_2_SO_4_, 2 g K_2_HPO_4_; 1 g NaCl; 0.4 g MgSO_4_·7H_2_O; 0.16 g EDTA-Na_2_; 0.01 g FeSO_4_·7H_2_O; and 1 mL of Trace elements. The Trace elements solution was prepared per liter with the following: 250 mg EDTA-Na_2_; 57 mg H_3_BO_3_; 110 mg ZnSO_4_·7H_2_O; 25.3 mg MnCl_2_·4H_2_O; 8.05 mg CoCl_2_·6H_2_O; 7.85 mg CuSO_4_·5H_2_O; and 5.5 mg Mo_7_O_24_ (NH_4_)_6_·4H_2_O. For cultivation, 150 mL Erlenmeyer flasks were filled with 50 mL of JM medium, inoculated with 10 mL of microalgae at a concentration of 0.1 g L^−1^, and covered with cotton caps. The cultures were illuminated continuously at room temperature using white fluorescent lamps (Model T8 36 W IP20, CMI, Munich, Germany) with a light intensity of 50 µmol m^−2^ s^−1^, measured by a luxmeter (Model HD 2302.0, Delta OHM, Padua, Italy). The inoculum was cultivated for around one week until the late exponential growth phase before being used for the experiments. Cheese whey (CW) samples were obtained from MAIL Industria Casearia, a dairy facility in Bellizzi, SA, Italy. The main chemical and physical parameters of the CW are detailed in Table 1. After collection, the CW was stored at 4 °C, filtered using glass microfiber filters (GF/CTM 47 mm diameter, Whatman, Incofar Srl, Modena, Italy) to remove solids, and sterilized at 121 °C and 0.1 MPa for 20 min before use in microalgae cultivation.

### 2.2. Cultivation Conditions and Experimental Setup

*Limnospira* was cultivated in 1L flasks (thereafter named PBRs) with a working volume of 600 mL. Each PBR was covered with a cotton cup, and filtered compressed air (containing 0.03% CO_2_ v v^−1^) was supplied using an air pump (GIS Air Compressor, Carpi, Italy). The PBRs were manually shaken daily at room temperature and exposed to a 12 h light/12 h dark photoperiod using white fluorescent lamps that provided a light intensity of 50 µmol m^−2^ s^−1^. Growth tests were conducted to assess cell growth, biomass production, and phicobiliproteins (PBPs). The detailed experimental setup can be found in Appendix A. Three types of CW were tested in the experiments: scotta whey (SW), buttermilk wastewater (BMW), and final dairy wastewater (DWW). For each type of CW, three different concentrations (1%, 2%, and 4%) were evaluated, with JM serving as the control. All tests were performed in triplicate over 18-day period. Microalgal growth was monitored by measuring optical density and biomass concentration. After cultivation, the final dry weight (g L^−1^) and PBP content (mg g^−1^_DW_) was determined. In all of the experiments, the initial inoculum concentration was set as 0.1 g L^−1^.

### 2.3. Cell Growth and Dry Weight Determination

The growth of *L. platensis* was tracked by measuring the absorbance (ABS) of the culture at 680 nm using a spectrophotometer (model ONDA V30 SCAN–UV VIS, ZetaLab, Padua, Italy). A regression equation correlating the dried biomass concentration with ABS was determined. The dry biomass concentration was assessed gravimetrically through the following steps: (a) a 10 mL sample of culture (V) was taken from the PBRs; (b) the sample was filtered through a pre-weighted (W_1_) glass microfiber filter (GF/C^TM^ 55 mm diameter, Whatman, Incofar Srl., Modena, Italy), and the biomass retained on the filter was dried at 105 °C overnight until a constant weight (W_2_) was achieved; (c) the filter paper had been previously dried in a forced-air oven (model 30, Memmert Gmbh, Scwabach, Germany) at 105 °C for 2 h, then cooled in a desiccator to room temperature, and weighed using an analytical balance (model M, Bel Engineering Srl, Monza, Italy).

The cell concentration (dry weight), X_dw_ (g L^−1^), was calculated using the following formula:X_dw_ = (W_2_ − W_1_)/V(1)
where W = weight (g) of dried algal biomass and V = volume (L) of the algae culture used for the test.

The average biomass productivity (∆X) was expressed as(∆X) = max X_dw_/t_max_(2)
where max X_max_ = maximum biomass (g L^−1^) obtained at (t_max_).

The specific growth rate (μ) was calculated according to the following equation:μ = (ln X_2_ − ln X_1_)/(t_2_ − t_1_)(3)
where X_2_ and X_1_ = dry biomass concentration (g L^−1^) at time t_2_ and t_1_, respectively.

The pH of culture suspensions was measured by a pHmeter (model HI 2210, Hanna Instruments, Woonsocket, RI, USA).

### 2.4. Phycobilinprotein Extraction and Spectrophotometric Determination

The extraction of PBPs was performed using an aqueous saline solution as described by Herrera et al. [36]. Specifically, 10 g of frozen *L. platensis* biomass was placed in 50 mL of an aqueous buffer solution containing 1% calcium chloride (10 g L^−1^) and subjected to repeated freezing and thawing steps until a complete cell disruption occurred. The mixture was stirred for 30 to 45 min. This extraction process was repeated twice, and the resulting phycobilin solution was separated by centrifugation at 8000 rpm for 10–15 min. The blue supernatant obtained was then used for optical measurement using a spectrophotometer. The concentration of different PBPs, including C-phycocyanin (PC), allophycocyanin (APC), and phycoerythrin (PE), was determined by measuring the absorbance of the extract at three specific wavelengths: 565 nm, 620 nm, and 650 nm.

The concentration of these PBPs, as mg mL^−1^ extract, was then determined from the equations established by Bryant et al. [37].(4)$$\left[\mathrm{PC}\right]=\frac{A_{620}-0.72\times A_{652}}{6.29}$$(5)$$\left[\mathrm{APC}\right]=\frac{A_{652}-0.191\times A_{620}}{5.79}$$(6)$$\left[\mathrm{PE}\right]=\frac{A_{565}-2.41\times \left[\mathrm{PC}\right]-1.40\times \left[\mathrm{APC}\right]}{13.02}$$

The concentration of total PBPs was determined as the sum of PE, PC, and APC in mg mL^−1^ of the extracted supernatant as follows:[PBPs] = [PC] + [APC] + [PE](7)

The extraction yield, estimated by relating the concentrations (expressed in terms of mg mL^−1^) to the biomass of *L. platensis* used (in terms of mg of dry weight), was obtained as follows:PBP = ([PBPs] (mg mL^−1^ of extract) × volume of extract))/wet biomass × 10%(8)

The pycocyanin (PC and APC) purity was calculated according to the following equations:PC Purity = A_620_/A_280_(9)APC Purity = A_650_/A_280_(10)

### 2.5. FAMEs and Healthy Parameters Determination

The fatty acids methyl ester (FAME) analysis followed the method outlined by Breuer et al. [38]. A detailed description of the protocol adopted can be retrieved in Russo et al. [39]. Briefly, lyophilized biomass (10 mg) was extracted using a methanol/chloroform mixture (4:5 v v^−1^) containing tritridecanoin (TAG 39:0, 13:0/13:0/13:0) as internal standard. The samples were vortexed, sonicated, and centrifuged, and the chloroform phase was collected. Lipids were trans-esterified using methanol with 5% sulfuric acid (v v^−1^) at 70 °C for 3 h. The resulting FAMEs were extracted with *n*-hexane, washed with water, and analyzed using gas chromatography–mass spectrometry (GC-MS). GC-MS was performed on Trace 1300 system (Thermo Fischer Scientific, Watham, MA, USA) equipped with a triple quadrupole mass spectrometer (TSQ 9000) (Thermo Fischer Scientific, Watham, MA, USA) and an Agilent HP-5 fused silica capillary column (Agilent Technology, Santa Clara, CA, USA). The injector temperature was 250 °C, and helium was used as the carrier gas. The oven temperature was programmed from 50 °C to 300 °C in multiple ramps. Peak identification was based on retention times compared to a Supelco 37 component FAME Mix (Sigma Aldrich, Darmstadt, Germany). The data are expressed as a mg g^−1^ of dry weight (mean ± standard deviation) and were calculated using the equation provided by Breuer et al. [38]:(11)$$\mathrm{FA}\;\left(\frac{\mathrm{mg}}{g}\right)=\mathrm{IS}\;\mathrm{added}\times \frac{\frac{\left(\mathrm{Area}\;\mathrm{of}\;\mathrm{individual}\;\mathrm{FAME}\right)}{\left(\mathrm{Area}\;\mathrm{of}\;C13:0\;\mathrm{FAME}\;\times \;\mathrm{Rel}.\;\mathrm{Resp}.\;\mathrm{Factor}\;\mathrm{individual}\;\mathrm{FAME}\right)}}{g\;\mathrm{of}\;\mathrm{biomass}\;\mathrm{added}}$$

The relative abundance of each FA was calculated by dividing the concentration of each FA by the total FA content.

Based on the unsaturated FAs, the atherogenic (AI), thrombogenic (TI), and hypocholesterolemic/hypercholesterolemic indexes (h/H) were calculated.

AI and TI were obtained using the formula proposed by Ulbricht et al. [40], as follows:AI = [C12:0 + 4] × [(C14:0 + C16:0]/[∑MUFA + ∑(*n* − 6) + (*n −* 3)](12)TI = [C14:0 + C16:0 + C18:0]/[0.5 × ∑MUFA + 0.5 × ∑(*n* − 6) + 3 × ∑(*n −* 3)] + (∑(*n* − 6)/∑(*n* − 3)](13)

The h/H ratio was calculated according to the equation suggested by Fernandez et al. [41]:h/H = [(Σ (C18:1n − 9, C18:1n − 7, C18:2n − 6, C18:3n − 6, C18:3n − 3, C20:3n − 6,C20:4n − 6, C20:5n − 3, C22:4n − 6, C22:5n − 3 and C22:6n − 3)/Σ (C14:0 and C16:0)](14)

### 2.6. Statistical Analysis

Each experimental condition was examined in triplicate. Statistical analysis on biomass, specific growth rate, and FAME profile was conducted using MetaboAnalysts 5.0 platform, developed by McGill University, Montreal, Canada. Differences between groups were assessed through one-way analysis of variance (ANOVA), followed by Tukey’s Honestly Significant Difference (HSD) test. Results were considered statistically significant at 95% confidence level, with a probability threshold of 0.05.

## 3. Results

### 3.1. Cheese Effluents Composition

Table 1 presents the fundamental physical/chemical characteristics of the three CWs utilized in this work: the scotta (SW) that is the effluent resulting from the production of ricotta, the buttermilk effluent (BMW) typically generated by butter or cream production, and the final dairy wastewater (DWW). It can be inferred how the loads of organic matter in terms of organic carbon were consistent in SW (BOD5 > 43 g L^−1^ and COD > 91 g L^−1^), this effluent being obtained from the first step of cheese making and therefore still retaining most of its original charge of organic matter. On the other hand, BMW and DWW, which represent subsequent steps of the dairy process, are characterized by a progressive pauperization of the carbon content. Similarly, the N and P content was also progressively reduced according to the extent of cheese effluent treatment. It should be considered that, despite cheese effluents possibly exhibiting different chemical compositions, based on the technological steps employed for manufacturing dairy products, these effluents are usually characterized by the presence of d-Lactose, soluble proteins, lipids, and salts able to sustain microalgae growth [33]. Their typical organic content, in terms of BOD and COD, can vary from 0.1 to 100 g L^−1^ [42]. As reference of their rich organic loads, 1 kg of lactose, protein, and fat corresponds to 1.13, 1, and 3 kg of COD, respectively [43]. A common procedure to allow microalgae growth inside a culture medium with a huge organic content, such that of a DWW, is its physical and chemical pre-treatment [44]. The decision to employ extremely low whey concentrations (cfr. Appendix A) was influenced by the elevated TOC content in this effluent.

### 3.2. Growth Profile and Biomass Composition of L. platensis Using CW

*L. platensis* was cultivated under both photoautotrophic and mixotrophic conditions using CW as an organic substrate. Three distinct cheese effluents, varying in their organic carbon and CW contents (refer to Table 1), were utilized for the experiments. *L. platensis* was grown in three different concentrations of each effluent, ranging from 1 to 4 v v^−1^, over a period of 18 days until it reached the early stationary growth phase. The objective of this study was to evaluate essential kinetic parameters, including maximum biomass concentration (*X_max_*), average biomass productivity (*∆X*), doubling time (*t_d_*), and specific growth rate (*µ*).

Figure 1 illustrates the growth curves in terms of the biomass concentration of *L. platensis* under both mixotrophic and photoautotrophic (CTRL) conditions. In all of the mixotrophic systems, a lag phase persisted at approximately 72 to 96 h, whereas no lag phase was evident in the CTRL conditions. A similar trend was reported in previous studies involving the cultivation of *L. platensis* [45] and of *Chlorella vulgaris* [46] using CW.

In our study, the extended adaptation phase was mainly attributed to the time needed by *L. platensis* to adjust to the new growth conditions represented by the addition of CW to the CTRL. It is important to note that the CW content in the three effluents varies based on the cheese processing stage, with the percentage of CW being the most stable in SW and gradually decreasing in BMW and DWW. Acclimation is a crucial stage in the adaptation of cyanobacteria and significantly influences the overall performance of the culture. Following this phase, the exponential growth phase across all samples lasted up to 14 days, displaying different growth patterns. Around the midpoint of the cultivation period (9 days), all of the systems investigated (except SW-4%, and BMW-4%) exhibited DWs higher than 1 g L^−1^. By the end of the cultivation period, the DWW-2% and DWW-4% systems surpassed the control, with the DWW-2% system continuing to show increasing DW values on the 18th day.

Figure 2 illustrates the four key kinetic parameters measured during the cultivation of *L. platensis* with the addition of scotta whey to JM. The maximum biomass concentration reached 3.30 g L^−1^ with SW-2%, which was significantly comparable to the control (CTRL) at 3.06 g L^−1^ and nearly double that of SW-1% (1.74 g L^−1^) and SW-4% (1.54 g L^−1^), these last two not being statistically different between them (Figure 2a).

In Figure 2b, only the specific growth rate for SW-4% (0.11 day) was significantly lower than that of the CTRL (0.27 day) and the other two SW systems. Furthermore, SW-2% exhibited the highest average biomass productivity at 235 mg L^−1^ day^−1^, representing over a 50% increase compared to SW-1% (105 mg L^−1^ day^−1^) and SW-4% (116 mg L^−1^ day^−1^), as shown in Figure 2c.

As the CW content decreased in BMW (see Figure 3) and more significantly in DWW (see Figure 4), *ΔX* showed significant consistent effects. Conversely, *X_max_* achieved higher values in BMW systems than those recorded with DWW. Utilizing buttermilk as an organic carbon source, the highest *X_max_* of 2.33 g L^−1^ was attained with BMW-2% (Figure 3a), which showed significantly higher results than BMW-4% but lower than BMW-1%, all three of the systems being significantly lower compared to the CTRL anyway. *µ* (in the range of 0.11 to 0.12 day) was not significantly different among the three systems but did significantly differ compared to the CTRL (Figure 3b). *ΔX* ranged from 117 to 128 mg L^−1^ day^−1^ (Figure 3c), with significant differences between them and also compared to the CTRL. A different trend compared to BMW was noted when *L. platensis* was cultivated in DWW, with the highest *X_max_* of 3.19 g L^−1^ achieved with DWW-2%. This value was not significantly different compared to the CTRL (3.06 g L^−1^) and to the other two systems, 2.87 g L^−1^ and 2.69 g L^−1^ for DWW-1% and DWW4%, respectively (Figure 4a). *µ* displayed a similar trend as observed in BMW, with the three BMW systems not significantly different amongst themselves but statistically lower compared to the CTRL (Figure 4b), while *ΔX* was significantly improved in DWW-2% (177 mg L^−1^ day^−1^) and DWW-4% (160 mg L^−1^ day^−1^) compared to the CTRL (157 mg L^−1^ day^−1^) (Figure 4c).

The impact of mixotrophc conditions on the biomass composition of *L. platensis* regarding macronutrients, such as total carbohydrates (TC), total proteins (TP), and total lipids (TP), is illustrated in Figure 5. The distinct characteristics of the three CW effluents, particularly in terms of organic load and composition, appear to significantly influence the TC component, while the effect on TL is less pronounced. Conversely, variations in TP among the three mixotrophic systems are more consistent. TP constitutes the largest fraction, followed by TL and TC, in both photoautotrophic and mixotrophic conditions. Specifically, under mixotrophy, TP ranged from 22% in DWW-2% to 33.63% in SW-2%, compared to the CTRL at 26.80%. Regarding the TL component, *L. platensis* grown in SW and BMW systems demonstrated higher TL values compared to the CTRL and DWW systems. Notably, BMW-1% and BMW-2% recorded TL values of 28.79% and 26.89%, respectively, while TL in SW systems ranged from 23.58% to 26.55%, compared to the CTRL value of 14.63%.

The distribution of TC displayed a contrasting trend across the three mixotrophic systems, with values ranging from 6.95% to 33.41%, but following different patterns. In SW, the highest TC content was observed with 4% of the CW (29.11%), whereas in BMW and DWW, the peak occurred with 4% CW (33.41%) and with 2% CW (28.46%). In all three mixotrophic systems, the lowest TC values were higher than the control, with SW-1% only slightly higher (6.95%), and BMW-2% (15.53%) and DWW-4% (14.50%) three times higher compared to the control (5.31%).

This macronutrient distribution, particularly the elevated lipid fraction and the low carbohydrate fraction for the SW systems, deviates significantly from the typical chemical composition of *L. platensis*, which generally comprises 15–25% carbohydrates, 55–70% proteins, and 4–7% lipids, as reported by Markou et al. [47]. Growth conditions, whether batch or continuous, affect not only microalgae growth and biomass productivity but also their biochemical composition [48]. Under mixotrophic conditions, compared to photoautotrophy, a substantial shift in protein and lipid compositions was observed. The addition of CW from three different sources led to an increase in terms of TC, TP, and TL, albeit to a different extent and with different behavior based on the organic source. SW and BWM produced an increase in all three components compared to the control. Conversely, the inclusion of CW in DWW caused a notable increase in TC, but slightly reduced TP fraction and increased TL fraction, respectively.

### 3.3. Phycobiliprotein Production by L. platensis Under Mixotrophic Conditions

Figure 6 illustrates the concentration of PC, APC, PE, and total phycobiliproteins (PBPs) in extracts from the *L. platensis* biomass grown under both photoautotrophic and mixotrophic conditions. The mixotrophic cultures were supplemented with cheese whey (SW), buttermilk (BMW), and dairy wastewater (DWW).

The data showed that pigment concentrations were generally higher under photoautotrophic conditions, while in mixotrophic cultivation, the results varied depending on the type of dairy effluent used. Among the effluents, SW produced the lowest pigment levels, with PBPs ranging from 0.47 to 0.60 mg L^−1^ and PC from 0.34 to 0.46 mg L^−1^. In contrast, BMW-1% led to a significant increase in PC production, reaching 0.80 mg L^−1^—more than double the values observed with SW. A similar pattern was observed for PBRs in these conditions. As for APC and PE, the highest concentrations were achieved using BMW-1% and DWW-4%, with values of 0.29 mg L^−1^ and with BMW-1% with a value of 0.16 mg L^−1^, respectively, compared to the control values of 0.32 mg L^−1^ and 0.17 mg L^−1^.

Previous studies indicate that PC production is driven by a complex interplay of factors, including the composition of the growth medium, the presence of organic carbon sources, and the physiological responses of microalgae to specific culture conditions (such as an appropriate addition of a N source), which create a stress environment conducive to PC synthesis [49]. However, in this study, the mixotrophic cultures did not exhibit a notable rise in total PBP concentrations compared to the photoautotrophic control. This outcome contrasts with earlier observations in *L. platensis* grown mixotrophically using CW as an organic carbon source, where a positive correlation between organic load and PBP content was demonstrated in larger-scale processes [23]. Similarly, higher PC levels compared to those observed in the conventional control Zarrouk medium were reported for *L. platensis* cultivated in tofu WW under mixotrophic conditions [50] and for *Galdieria sulphuraria* grown in media containing buttermilk [34].

In the current study, PC purity under mixotrophic conditions was found to have an EP ranging from 0.2 to 0.4 with SW, from 0.2 to 0.45 with BMW, and from 0.45 to 0.70 with DWW, compared to 0.55 in the control (Figure 7a). The highest level of purity (0.70) was reported by DWW-4%. Correspondingly, the PC yields ranged from 20 to 24 mg g^−1^ for SW, from 18 to 50 mg g^−1^ for BMW, and from 20 to 48 mg g^−1^ for DWW cultures compared to 49 mg g^−1^ in the control (Figure 7b). The highest yields, 50 mg g^−1^ and 48 mg g^−1^, were exhibited by BMW-1% and DWW-4%, respectively. These findings are in part consistent with recent research carried out by Russo et al. [39] and by Cavallini et al. [23], which demonstrate that an organic source such as dairy WW in low concentrations (0.5–2% v v^−1^) can enhance PC synthesis in mixotrophic cultures of *L. platensis*.

### 3.4. FAME Profile by L. platensis Under Mixotrophy

The fatty acid methyl ester (FAME) composition of L. platensis grown under mixotrophic conditions using three different CW sources, in comparison to the control (CTRL), is presented in Table 2. No notable differences were observed in the FAME profile between the mixotrophic systems, although significant variations emerged when compared with the photoautotrophic system. Specifically, higher concentrations of myristic acid (C14:0), palmitic acid (C16:0), hexadecenoic acid (C16:1), elaidic acid (C18:1 trans), oleic acid (C18:1 cis), linoleic acid (C18:2), and y-linolenic acid n-6 (C18:3) were found under mixotrophy, whereas heptadecanoic acid (C17:0), stearic acid (C18:0), α-Linolenic acid (C18:3), and 8,11,14-eicosatrienoic acid (C20:3) were lower.

In the control group (CTRL), C16:0 was the most dominant FA at 40.09%, followed by C18:0 (26.81%), C18:3 n-6 (10.60%), and C18:2 (7.26%). Similarly, in the SW group, C16:0 ranged from 40.35% to 42.77%, followed by C18:0 (15.53–19.71%), C18:3 n-6 (13.33–15.66%), and C18:2 (8.40–9.18%). In the BMW group, C16:0 was also the most prevalent at 40.88–41.81%, trailed by C18:0 (14.99–18.56%), C18:3 n-6 (14.48–15.70%), and C18:2 (8.58–9.75%). Lastly, the DWW group exhibited a similar hierarchy, with C16:0 (41.83–42.25%) leading, followed by C18:0 (19.63–22.35%), C18:3 n-6 (12.63–13.65%), and C18:2 (7.55–8.39%).

C16:0 emerged as the dominant FA across all cultivation systems, and the total percentage of C16-C18 FAs in *L. platensis* showed only slight variation between photoautotrophic (96.33%) and mixotrophic conditions, which ranged from 95.33% in DWW-1% to 96.41% in SW-1%. These results align with Cavallini et al. [23], who reported a similar FA distribution pattern in *L. platensis* grown under both autotrophic and mixotrophic conditions with CW supplementation. Similarly, Russo et al. [39] found comparable trends when cultivating *L. platensis* in autotrophic and mixotrophic systems using brewery WW under salt stress induced by seawater addition. This study also highlighted notable changes in saturated (SFA), monounsaturated (MUFA), and polyunsaturated fatty acid (PUFA) levels between photoautotrophy (CTRL) and mixotrophy. Additionally, the mixotrophic response varied based on CW concentration. Specifically, a gradual increase in CW concentration in SW resulted in higher UFAs and a concomitant decline in SFAs compared to the CTRL. Conversely, the BMW group exhibited the opposite trend, with an increase in SFAs and a drop in UFAs. The DWW group partially mirrored the SW trend, though the increase in UFAs and decrease in SFAs did not scale directly with CW concentrations. The highest percentage of polyunsaturated fatty acids (PUFAs) was observed in BMW-1% (27.32%), while SW-4% had the highest proportion of monounsaturated fatty acids (MUFAs) (16.57%).

The Thrombogenicity Index (TI), Atherogenicity Index (AI), and hypocholesterolemic/hypercholesterolemic (h/H) ratios were calculated from the FA profile to assess the nutritional index and the potential health benefits of *L. platensis* compared to those of other microalgae grown under different organic sources and conditions (Table 3). These three parameters are critical in evaluating the potential cardiovascular health impacts of microalgae, particularly in food and nutraceutical applications. In particular, the h/H ratio is a crucial indicator in cholesterol metabolism, with higher h/H values considered more beneficial for cardiovascular health. These values are thought to provide a clearer reflection of the potential impact on cardiovascular disease risk. In our study, *L. platensis* exhibited an h/H ratio of 0.64 under photoautotrophic conditions, while the ratios under mixotrophic conditions with the addition of CW to the JM medium ranged from 0.62 to 0.81. The highest value (0.81) was observed with the use of scotta (SW-4%) (Table 2). The h/H ratios under mixotrophic conditions with SW (0.74–0.81), BMW (0.69–0.80), and DWW-1% (0.69) were higher than the values (0.60–0.66) reported for *L. platensis* by other researchers under photoautotrophy (Table 3). However, several freshwater microalgae strains reported in this table exhibited a wide range of h/H values when grown under mixotrophic conditions, ranging from 0.74 to 4.22. To interpret this wide variability between different microalgae, it is important to note that differences in oil extraction methods (including the types of solvents used) from microalgae cells are not consistently reported in the literature.

The highest values of 0.86 and 0.97 for the AI and TI, respectively, were obtained for *L. platensis* grown in SW-4%. The wide variability in terms of AI ad TI values exhibited by *L. platensis* compared to the other microalgae can be explained considering that these microalgae belong to different phyla (Bacillariophyta, Cyanobacteria, and Ochrophyta) and that the different origin of the organic source used in the culture medium (brewery, dairy, molasses, glucose) may have a significant impact on the enzymatic apparatus involved in the FA metabolism.

## 4. Discussion

Dairy byproducts like cheese whey (CW), buttermilk, and dairy effluents vary in composition, containing sugars, organic acids, and fats. CW, also known as ricotta cheese or scotta, is a thin and watery white to yellow/green opalescent liquid obtained during the cheese-making process by coagulating and separating casein proteins from milk [59]. It comprises roughly 55% milk nutrients and is rich in organic matter, exhibiting substantial potential for mixotrophic and heterotrophic microalgae cultivation. Lactose dominates CW solids (75%), complemented by galactose, oligosaccharides, lactic acid and acetate, and minor proteins such as β-lactoglobulin and α-lactalbumin, which have high nutritional value [28,60]. Buttermilk has a similar composition but contains more fats and fewer organic acids due to fermentation processes. While cyanobacteria, including *L. platensis*, cannot hydrolyze lactose directly, they can assimilate acetate and glycerol, enhancing biomass and protein production under specific conditions, especially in mixotrophic or heterotrophic growth modes [61,62]. Historically, CW was considered waste, posing environmental challenges due to its high levels of biochemical and chemical oxygen demand (BOD and COD) compared to urban WW [60]. It was commonly discarded or used as a fertilizer, but its potential as a substrate for *L. platensis*, particularly using glycerol and acetate, offers a sustainable alternative.

Mixotrophic cultures grow faster than photoautotrophic and heterotrophic ones due to their capacity to utilize multiple growth substrates while performing photosynthesis, which preserves the acetyl-CoA pool for CO_2_ fixation via the Calvin cycle and the synthesis of extracellular organic carbon [63]. It should be considered that the role of light in *A. platensis* growth within dairy residues was not explicitly tested in this study, as the focus was on evaluating mixotrophic cultivation under light conditions. While light likely supports photosynthetic activity, the presence of assimilable carbon sources, such as glycerol and acetate, may enable heterotrophic growth. Future studies should explore this by comparing growth under light and dark conditions to distinguish between mixotrophic and heterotrophic growth modes. Such investigations are critical for optimizing cultivation strategies and understanding *L. platensis* metabolic flexibility.

The potential of DWW to stimulate mixotrophic metabolism in microalgae and cyanobacteria has been investigated in various studies. Studies indicate that the ideal CW concentration for mixotrophic cultivation is 3.0% v v^−1^, with higher concentrations (5–100% v v^−1^) leading to growth inhibition [32]. Salla et al. [64] found that CW concentrations ranging from 1.25 to 2.5% supported *Limnospira* growth, while Pereira et al. [45] demonstrated that *S. platensis* thrived in Zarrouk’s medium supplemented with 2.5–10% CW, with 2.5% yielding the best results. Athanasiadou et al. [64] achieved the highest biomass concentration (1.06 g L^−1^) under alternating light/dark conditions at 2.5% untreated CW. Similarly, Miotti et al. reported that *Chlorella vulgaris* grown in DWW containing different glycerol concentrations under mixotrophic conditions produced a significantly greater biomass yield (1.72 g L^−1^) than autotrophic growth (1.08 g L^−1^) [24]. These findings underscore the value of DWW and CW as substrates for mixotrophic microalgae cultivation, enhancing both growth and lipid productivity.

In our study, *L. platensis* exhibited the highest biomass yields under photoautrophic conditions, except for SW-2% and DWW-2%. This may be due to lactose, the main sugar in scotta, buttermilk, and dairy products, being a disaccharide that *L. platensis* cannot directly metabolize due to the lack of necessary enzymes [47]. Unlike some *Chlorella* species, *L. platensis* relies on simpler sugars like glucose and sucrose, as well as proteins and vitamins, for growth [45]. Furthermore, analysis of the N:P molar ratio in CW revealed it to be significantly lower than the approximately 5:1 ratio reported for DWW by Gramegna et al. [65] and for CW by Kiani et al. [66]. Additionally, this ratio falls below the Redfield ratio (N:P of 16:1), indicating that CW acts as a N-limited medium for microalgae growth.

The composition of *L. platensis* cultivated under mixotrophic conditions with CW indicates considerable variations in TC, TP, and TL across different dilution ratios. Higher CW dilution ratios, such as SW-4% and BMW-4%, led to increased TC, likely due to enhanced nutrient availability. This observation aligns with studies showing that elevated levels of dairy substrates can boost pigment production, although there can be a decrease in protein, as seen in the DWW1-4% treatments. *Desmodesmus* sp. with 15% CW and 50% Bold’s basal medium showed significant improvements in growth (303%), productivity (325%), lipids (3.89%), and carbohydrates (1.95%) [67]. Similarly, Salati et al. [68] demonstrated that mixotrophic cultivation of *Chlorella* using agro-food byproducts like CW enhances algal production, particularly protein yield. Other microalgae, such as *Tetradesmus obliquus* and *Cyanothece* sp., maintained stable protein levels across various CW concentrations of 0.5–4.5% [32]. At 3.5% CW (v v^−1^), *T. obliquus* achieved productivities of 48.69, 20.64, 7.02, and 10.97 mg L^−1^ day^−1^ for biomass, lipid, carbohydrates, and protein, respectively. Meanwhile, *Cyanothece* produced 52.78 mg L^−1^ day^−1^ of biomass, 11.42 mg L^−1^ day^−1^ of lipids, 4.31 mg L^−1^ day^−1^ of carbohydrates, and 7.89 mg L^−1^ day^−1^ of protein at 4.5% CW (v v^−1^). Youssef et al. [32] highlight the potential of dairy byproducts as nutrient sources for maximizing bioactive compound production depending on species and dilution ratios.

The suitability of PC for various uses is influenced by its level of purity, assessed through the absorbance ratio A620/A280, referred to as extraction purity (EP). If the EP is 0.7 or higher, as is the case in this study, the PC is considered food grade, making it suitable for use as a food additive or a natural blue colorant in cosmetics. When the EP falls between 0.7 and 3.9, it is classified as reagent grade, with EP values of 1.5 or more being appropriate for cosmetic applications. An EP of 4 or higher qualifies PC as analytical grade, suitable for pharmaceutical applications [69]. The purity is strongly influenced by extraction techniques involving factors such as temperature, pH, solvent type, biomass-to-solvent ratio, and whether the biomass is dried or fresh. The commercial value of PC is highly dependent on its purity level [70], with analytical-grade PC priced at 4,500 US$ g^−1^, for high-purity applications such as pharmaceuticals, therapeutic, biomedicine, and cosmetics [71], while lower-purity PC is used in commercial food products or as a biocolorant [72]. These high prices are primarily due to the challenges involved in the extraction and purification processes, making PC an expensive protein pigment [20]. The global PC market is projected to grow to $245.5 million by 2027 and $279.6 million by 2030 [73], reflecting its growing demand across industries like food, cosmetics, and pharmaceuticals.

Figure 7 illustrates the effect of varying concentrations of SW, BMW, and DWW on the purity and yield of C-PC. The relatively low C-PC purity in SW-treated groups (1% and 2%) suggests that the introduction of CW may have increased the turbidity of the medium. This turbidity reduces light penetration into the culture [74], limiting photosynthetic efficiency and, consequently, C-PC production. The low C-PC yield observed under SW conditions further reinforces the idea that its nutrient profile and opacity do not create the stress or nutrient dynamics necessary to boost pigment synthesis [50]. Lower light availability limits the photosynthetic activity needed for C-PC accumulation. BMW-treated groups also exhibited modest C-PC purity, but yields vary across concentrations. The 2% BMW condition shows a relatively high C-PC yield, potentially because moderate nutrient levels provide adequate support for growth without overwhelming the system. However, higher BMW concentrations (4%) may introduce excessive organic loading or nutrient oversaturation, reducing the physiological stress required to trigger accessory pigment production, like C-PC, which is typically downregulated under stress conditions like nutrient limitation [75]. DWW-treated groups, especially at 4%, show the highest C-PC purity and yield. This may result from a balanced interaction between nutrient availability and stress. DWW likely contains a mix of organic carbon, N, and other micronutrients that, at higher concentrations, induce mild stress, such as nutrient fluctuations or oxidative stress. These stressors enhance the production of secondary metabolites, including C-PC. The observed higher yields under DWW treatments suggests that their nutrient composition combined with moderate nutrient stress could boost C-PC output [76]. For instance, oxidative stress from organic matter degradation can stimulate the production of protective and photosynthetic pigments, boosting overall output. Overall, variations in C-PC purity and yield across SW, BMW, and DWW concentrations highlight the role of environmental stressors, nutrient composition, and light availability in driving microalgal responses [77]. Higher turbidity in SW and BMW reduces light availability, lowering C-PC production. Meanwhile, DWW, particularly at 4%, creates an ideal stress environment that optimizes both purity and yield. This study provides valuable insights for developing algal cultivation strategies to optimize bioactive compound production while supporting sustainable and eco-friendly practices.

C16:0 is a crucial energy source in infant nutrition, comprising 20–30% of breast milk, but elevated levels of free SFAs, particularly C16:0 and C18:0, in adults are associated with cardiovascular disease due to oxidative stress and vascular endothelial dysfunctions [78,79]. Maintaining normal levels of these SFAs is critical in avoiding such issues [80]. As can be observed in Table 2, C16 levels remained almost unchanged across all mixotrophic conditions, while C18 levels were considerably decreased under mixotrophy.

Olive oil, rich in beneficial MUFAs like oleic acid (C18:1, 70–80% of its FA content), mitigates cardiovascular risk [81]. In this study, the C18:1 content in *L. platensis* increased under mixotrophic conditions, especially with scotta, buttermilk, and dairy effluent at 1%. However, mixotrophy also increased SFA expression over PUFAs and MUFAs. 

α-linolenic acid (ALA, C18:3 *ω*-3) and γ-linolenic acid (GLA, C18:3 *ω*-6) are two PUFAs commonly present in oil derived from microalgae and cyanobacteria [82]. GLA, the primary isomer of this FA in *L. platensis* [83], was enhanced under mixotrophic conditions, particularly with buttermilk at 1% (15.70%) and 2% (15.34%), and scotta at 2% (15.66%), compared to the photoautotrophic control (10.60%). A recent review investigated the role of temperature, light intensity, N cell concentration, growth phase, and light/dark cycles, in promoting lipids and GLA synthesis in *Limnospira* [84]. In contrast, ALA decreased significantly with CW addition compared to the CTRL (1.24%). ALA serves a as a precursor for the synthesis of eicosapentaenoic acid (EPA, C20:5 ω-3) and docosahexaenoic acid (DHA, C22:6 ω-3). ALA must be obtained through the diet because the human body cannot synthesize it. The body converts ALA into EPA and DHA, which are vital for maintaining the proper function of key organs. However, this conversion is relatively inefficient, with about 5% to 10% of consumed ALA being converted into EPA, and roughly 1% of this EPA being further converted into DHA [85].

The PUFA to SFA ratio is a key nutritional indicator for cardiovascular health. PUFAs are known to reduce low-density lipoprotein cholesterol (LDL-C) and total serum cholesterol (making a higher PUFA ratio beneficial), while SFAs tend to elevate cholesterol levels [86]. The British Department of Health recommends a PUFA ratio above 0.45, and the WHO/FAO guidelines suggest maintaining a PUFA ratio above 0.4 to reduce chronic disease risk [87]. In this study, four out of nine mixotrophic systems with *L. platensis* achieved a PUFA ratio exceeding 0.4: SW-4% (0.42), SW-2% (0.43), BMW-2% (0.44), and BMW-1% (0.46), compared to the CTRL (0.29).

The TI varies significantly across microalgae species and cultivation methods. *Tetradesmus dimorphus* (formerly *Scenedesmus dimorphus*) grown on glucose shows the highest TI (4.0), indicating a higher thrombosis risk and potential cardiovascular concerns. In contrast, *Tribonema aequale* (Xanthophyceae) grown on glucose has a notably low TI (0.183), suggesting reduced clotting risk and greater cardiovascular health. For *L. platensis*, the organic source affects TI. Brewery residues yield a high TI (3.51), while dairy sources result in much lower values, such as 0.86 in this study (Table 3). This underscores the impact that cultivation media can have on the thrombogenic potential of microalgae, suggesting that dairy byproducts are particularly indicated in promoting the production of beneficial FAs for overall human health [88].

Similarly, the AI, indicating a food’s potential to cause arterial fat buildup, varies significantly. Higher AI values, like in *Tetradesmus dimorphus* (formerly *Scenedesmus dimorphus*) (1.68) and *Parachlorella kessleri* (formerly *Chlorella kessleri*) (1.64), suggest moderate atherosclerosis risk. On the other hand, lower AI values in *Chromochloris zofingensis* (Chlorophyta) (0.23 on molasses and 0.215 on dairy) indicate better cardiovascular health. Once again, the organic source used in cultivation has a significant influence on these health-related indices [88].

The h/H ratio reflects the nutritional quality of microalgae, particularly in terms of FA composition [89]. A higher ratio indicates a greater proportion of heart-healthy UFAs [90]. *Chlorella vulgaris* shows particularly high h/H ratios (2.67 and 2.8 on molasses and glucose, respectively), suggesting a high UFA content suitable for functional foods or dietary supplements. In contrast, *L. platensis* generally shows lower ratios (0.6-1.07, depending on cultivation), indicating a less beneficial FA profile.

Overall, Table 3 highlights how microalgae’s cardiovascular health potential varies significantly by species and cultivation conditions. Microalgae with high h/H ratios and low TI and AI values show the most promise for use in food and nutraceuticals aimed at improving heart health. Conversely, those with higher TIs or AIs may need tailored cultivation to optimize their nutritional profiles. This underscores the importance of selecting appropriate cultivation methods to enhance both the health benefits and safety of microalgae for diverse applications.

While promising results were observed under our laboratory conditions, the scalability of cultivating *L. platensis* using dairy residues requires careful evaluation. Variability in the composition of dairy residues, influenced by processing methods and seasonal factors, poses challenges for standardization. Additionally, pre-treatment steps may be needed to ensure consistent nutrient profiles, potentially increasing costs. Economic feasibility is a key factor for large-scale implementation. Utilizing dairy residues could reduce waste management costs and provide an affordable nutrient source, but the economic viability depends on achieving high biomass yields in outdoor or large-scale systems, where environmental conditions are less controlled. Optimizing cultivation parameters such as residue concentration, light intensity, and mixing will be essential to improve productivity and cost-effectiveness.

## 5. Conclusions

This study demonstrates the efficacy of using dairy byproducts, including scotta whey (SW), buttermilk wastewater (BMW), and dairy wastewater (DWW), as substrates for the mixotrophic cultivation of *Limnospira platensis*. The results showed that a 2% (v v^−1^) concentration of SW and DWW enhanced biomass production, achieving maximum concentrations of 3.30 g L^−1^ and 3.19 g L^−1^, respectively, compared to the control condition (3.06 g L^−1^). In terms of growth kinetics, *L. platensis* cultivated in SW-2% exhibited the highest average biomass productivity of 235 mg L^−1^ d^−1^ and a specific growth rate (μ) of 0.21 d^−1^, compared to 0.27 d^−1^ in the control.

Phycocyanin production was also enhanced under mixotrophic conditions, particularly in the BMW-1% treatment, which yielded 50 mg g^−1^ of dry weight, approaching the control’s 49 mg g^−1^. Moreover, the highest phycocyanin purity was achieved in DWW-4% cultures, with an extraction purity (EP) of 0.70, making it suitable for food-grade applications.

The FAME profiles showed consistent dominance of hexadecanoic acid (C16:0), ranging from 40.35% to 42.77%, across all mixotrophic conditions, similar to the control (40.09%). However, there were notable increases in PUFAs under mixotrophy, with the highest PUFA content of 27.32% recorded in BMW-1%. Additionally, the hypocholesterolemic/hypercholesterolemic (h/H) ratio improved under mixotrophic conditions, reaching a peak of 0.81 in SW-4%, compared to 0.64 in the control, indicating potential cardiovascular health benefits.

These findings underscore the potential of integrating dairy effluents into *L. platensis* production systems, offering a sustainable approach to both waste management and the generation of nutritionally and economically valuable biomass. This approach aligns with the goals of a circular bioeconomy, offering a cost-effective and environmentally friendly alternative to traditional cultivation methods.

## Figures and Tables

**Figure 1 life-15-00184-f001:**
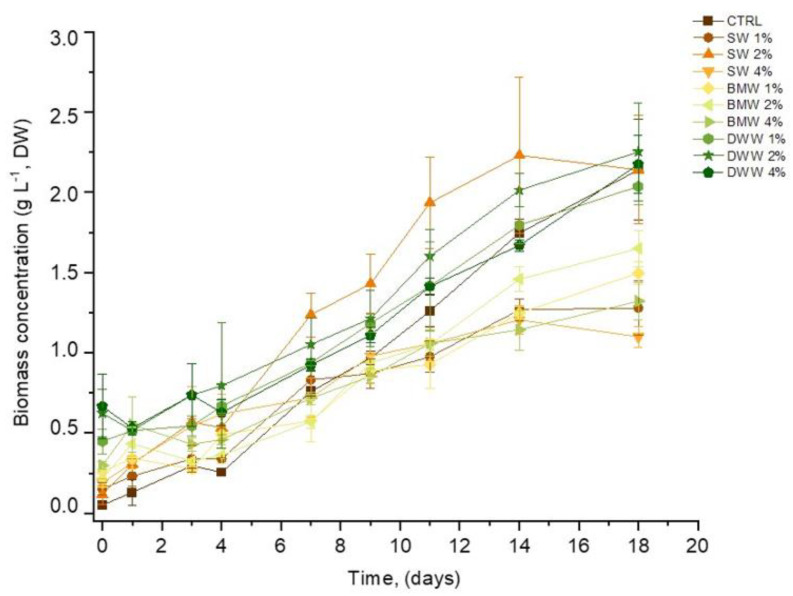
Time evolution of growth curves in terms of dry weight (g L^−1^) in SW, BMW, and DWW media containing different percentages of CW.

**Figure 2 life-15-00184-f002:**
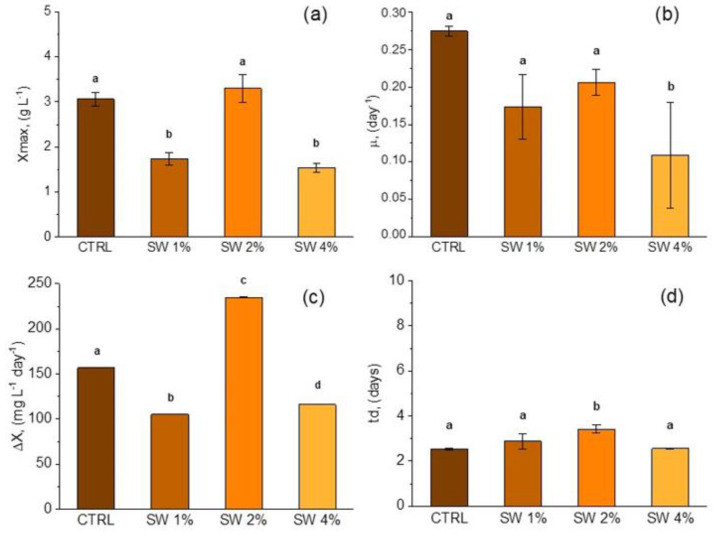
Comparison of growth performance indicators for maximum biomass concentration (*X_max_*) (**a**), specific growth rate (*µ*) (**b**), average biomass productivity (*ΔX*) (**c**), and doubling time (*t_d_*) (**d**) in SW media with different CW percentages. Mean differences were compared using ordinary one-way ANOVA followed by Tukey’s multiple comparisons test. For each trait, single-degree-of-freedom contrasts were applied to compare mean values (*n* = 3 ± SD) between control group and treatment groups, as well as among treatment groups. Means denoted by same letter are not significantly different at *p* ≤ 0.05, whereas different letters indicate significant differences at confidence level of at least 95% according to Tukey’s multiple comparisons test.

**Figure 3 life-15-00184-f003:**
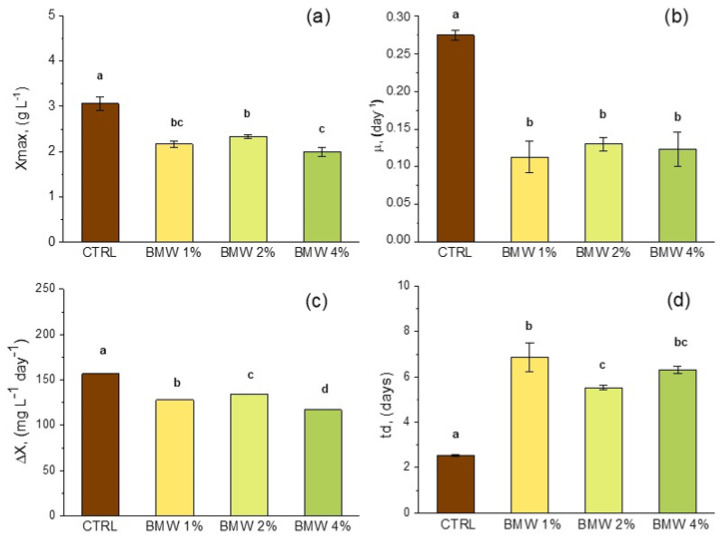
Comparison of growth performance indicators for maximum biomass concentration (*X_max_*) (**a**), specific growth rate (*µ*) (**b**), average biomass productivity (Δ*X*) (**c**), and doubling time (*t_d_*) (**d**) in BMW media with different CW percentages. Mean differences were compared using ordinary one-way ANOVA followed by Tukey’s multiple comparisons test. For each trait, single-degree-of-freedom contrasts were applied to compare mean values (*n* = 3 ± SD) between control group and treatment groups, as well as among treatment groups. Means denoted by same letter are not significantly different at *p* ≤ 0.05, whereas different letters indicate significant differences at confidence level of at least 95% according to Tukey’s multiple comparisons test.

**Figure 4 life-15-00184-f004:**
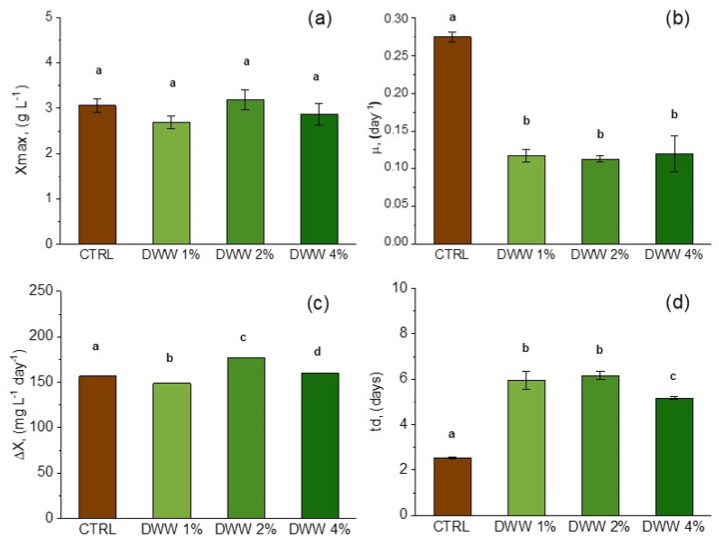
Comparison of growth performance indicators for maximum biomass concentration (*X_max_*) (**a**), specific growth rate (*µ*) (**b**), average biomass productivity (Δ*X*) (**c**), and doubling time (*t_d_*) (**d**) in DWW media with different CW percentages. Mean differences were compared using ordinary one-way ANOVA followed by Tukey’s multiple comparisons test. For each trait, single-degree-of-freedom contrasts were applied to compare mean values (*n* = 3 ± SD) between control group and treatment groups, as well as among treatment groups. Means denoted by same letter are not significantly different at *p* ≤ 0.05, whereas different letters indicate significant differences at confidence level of at least 95% according to Tukey’s multiple comparisons test.

**Figure 5 life-15-00184-f005:**
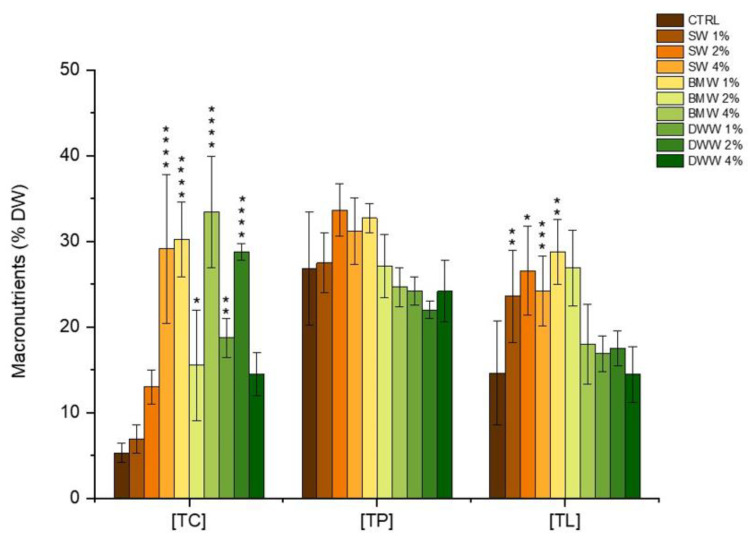
Total carbohydrates (TC), total lipids (TL), and total proteins (TP) obtained in SW, BMW, and DWW media under three different CW contents. Mean differences were compared using Tukey’s test (*n* = 3; * *p* indicates < 0.1; ** *p* indicates < 0.01; *** *p* indicates < 0.001; **** indicates *p* < 0.0001).

**Figure 6 life-15-00184-f006:**
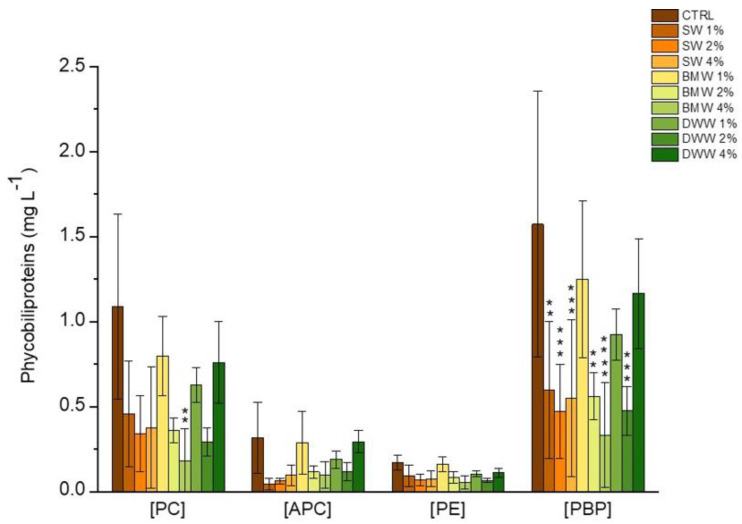
Concentration of phycocyanin (PC), allophycocyanin (APC), phycoerythrin (PE), and total phycobiliproteins (PBPs) in extracts of *L. platensis* grown in SW (a), BMW (b), and DWW (c) media under three different CW contents. Mean differences were compared using Tukey’s test (*n* = 3, ** *p* indicates < 0.01; *** *p* indicates < 0.001; **** indicates *p* < 0.0001).

**Figure 7 life-15-00184-f007:**
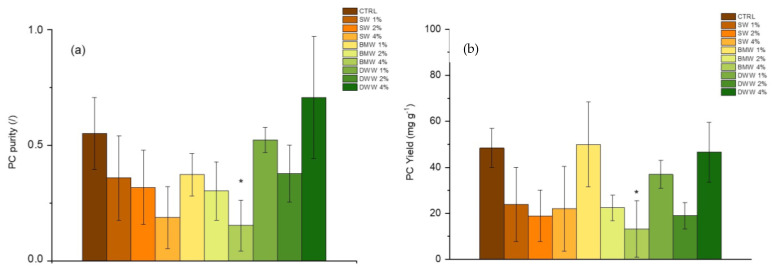
Purity (**a**) and yield (**b**) of phycocyanin (PC) in extracts of *L. platensis* grown in SW, BMW, and DWW media under three different CW contents. Mean differences were compared using Tukey’s test (*n* = 3, * *p* < 0.05).

**Table 1 life-15-00184-t001:** Composition of cheese whey used in this study.

Parameter	SW	BMW	DWW
BOD5	42,700	21,600	1246
COD	90,918	62,704	1528
TN	567	148	74
TP	575	235	16
pH	3.8	5.2	6.4

Note: SW = scotta whey, the remaining liquid after the production of ricotta; BMW = buttermilk wastewater; DWW = final cheese whey wastewater; BOD5 = biological oxygen demand; COD = chemical oxygen demand. All of the concentrations are expressed in terms of mg L^−1^.

**Table 2 life-15-00184-t002:** Impact of various growth media on FAME composition. Data are presented as mean% ± standard deviation (*n* = 6). The percentages are the total dry weights of the FAMEs.

FAMEs	C:N ^$^	CTRL	SW-1%	SW-2%	SW-4%	BMW-1%	BMW-2%	BMW-4%	DWW-1%	DWW-2%	DWW-4%
Myristic acid	14:00	1.80 ± 0.16 ^a^	1.96 ± 1.07 ^a^	2.32 ± 0.87 ^a^	2.60 ± 0.74 ^a^	2.57 ± 0.34 ^a^	2.58 ± 0.44 ^a^	2,16 ± 0.18 ^a^	2.90 ± 0.20 ^a^	2.74 ± 0.32 ^a^	2.72 ± 0.11 ^a^
Hexadecanoic acid	16:00	40.09 ± 4.77 ^a^	42.77 ± 1.54 ^a^	41.74 ± 1.07 ^a^	40.35 ± 1.11 ^a^	40.88 ± 0.58 ^b^	41.48 ± 0.50^ab^	41.81 ± 0.64 ^b^	41.83 ± 0.56 ^a^	42.55 ± 0.53 ^a^	42.15 ± 0.45 ^a^
Hexadecenoic acid	16:01	3.79 ± 0.70 ^a^	4.56 ± 0.71 ^a^	4.36 ± 0.46 ^a^	5.65 ± 1.00 ^a^	6.00 ± 1.29 ^a^	5.28 ± 0.38 ^a^	6.43 ± 0.22 ^a^	4.07 ± 0.24 ^a^	3.77 ± 0.37 ^a^	4.08 ± 0.91 ^a^
Heptadecanoic acid	17:00	0.22 ± 0.08 ^a^	0.18 ± 0.07 ^a^	0.15 ± 0.03 ^a^	0.20 ± 0.03 ^a^	0.14 ± 0.02 ^a^	0.15 ± 0.01 ^a^	0.14 ± 0.00 ^a^	0.13 ± 0.02 ^a^	0.12 ± 0.01 ^a^	0.12 ± 0.02 ^a^
10-Heptadecenoic acid	17:1 cis	0.27 ± 0.04 ^a^	0.31 ± 0.14 ^a^	0.31 ± 0.03 ^a^	0.36 ± 0.15 ^a^	0.28 ± 0.01 ^a^	0.25 ± 0.06 ^a^	0.20 ± 0.01 ^a^	0.22 ± 0.02 ^a^	0.17 ± 0.07 ^a^	0.21 ± 0.05 ^a^
Stearic acid	18:00	26.81 ± 2.35 ^a^	19.71 ± 7.37 ^ab^	17.11 ± 2.31 ^b^	15.53 ± 4.34 ^b^	14.99 ± 0.33 ^b^	16.18 ± 2.11 ^b^	18.56 ± 1.62 ^ab^	19.63 ± 0.15 ^a^	22.35 ± 1.71 ^a^	20.93 ± 1.97 ^a^
Elaidic acid	18:1 trans	0.63 ± 0.24 ^a^	0.78 ± 0.08 ^a^	1.54 ± 0.37 ^a^	3.80 ± 1.17 ^b^	1.37 ± 0.27 ^a^	1.13 ± 0.32 ^a^	1.53 ± 0.18 ^a^	1.21 ± 0.10 ^a^	1.23 ± 0.07 ^a^	0.87 ± 0.12 ^a^
Oleic acid	18:1 cis	5.91 ± 0.48 ^a^	6.29 ± 1.77 ^a^	5.59 ± 0.50 ^a^	6.19 ± 0.88 ^a^	6.05 ± 0.27 ^a^	5.98 ± 1.48 ^a^	4.63 ± 0.29 ^a^	6.18 ± 1.38 ^a^	5.00 ± 1.03 ^a^	5.97 ± 0.17 ^a^
Linoleic acid	18:02	7.26 ± 0.31 ^a^	8.40 ± 2.04 ^a^	8.82 ± 0.22 ^a^	9.18 ± 2.29 ^a^	9.75 ± 0.55 ^a^	9.45 ± 0.84 ^a^	8.58 ± 0.72 ^a^	8.39 ± 0.79 ^a^	7.55 ± 0.77 ^a^	8.06 ± 0.91 ^a^
α-Linolenic acid	18:3 *ω*-3	1.24 ± 0.05 ^a^	0.57 ± 0.61^a^	0.83 ± 0.74 ^a^	0.36 ± 0.22 ^a^	0.69 ± 0.42 ^ab^	0.56 ± 0.04 ^ab^	0.21 ± 0.09 ^b^	0.40 ± 0.06 ^a^	0.52 ± 0.15 ^a^	0.49 ± 0.28 ^a^
y-Linolenic acid	18:3 *ω*-6	10.60 ± 0.51 ^a^	13.33 ± 3.17^ab^	15.66 ± 1.12 ^b^	14.33 ± 2.63 ^ab^	15.70 ± 0.95 ^b^	15.34 ± 0.93 ^b^	14.48 ± 1.32 ^ab^	13.65 ± 0.71 ^a^	12.63 ± 1.02 ^a^	13.11 ± 1.07 ^a^
8,11,14-Eicosatrienoic acid	20:03	1.19 ± 0.12 ^a^	0.96 ± 0.30 ^a^	0.99 ± 0.25 ^a^	0.89 ± 0.23 ^a^	1.18 ± 0.23 ^a^	1.21 ± 0.19 ^a^	0.88 ± 0.04 ^a^	1.05 ± 0.23 ^a^	1.03 ± 0.20 ^a^	1.03 ± 0.15 ^a^
13-Docosenoic acid	22:01	0.19 ± 0.02 ^a^	0.19 ± 0.08 ^a^	0.58 ± 0.32 ^b^	0.57 ± 0.21 ^b^	0.40 ± 0.05 ^a^	0.41 ± 0.05 ^a^	0.39 ± 0.03 ^a^	0.34 ± 0.05 ^a^	0.35 ± 0.05 ^a^	0.24 ± 0.05 ^a^
Σ SFAs	/	68.92	64.62	61.32	58.68	59.78	60.39	62.67	64.49	67.76	65.92
Σ UFAs	/	31.08	35.39	38.68	41.33	41.02	39.61	37.33	35.51	32.25	34.06
Σ MUFAs	/	10.79	12.13	12.38	16.57	13.7	13.05	13.18	12.02	10.52	11.37
Σ PUFAs	/	20.29	23.26	26.30	24.76	27.32	26.56	24.15	23.49	21.73	22.69
PUFA:SFA	/	0.29	0.36	0.43	0.42	0.46	0.44	0.38	0.36	0.32	0.34
C16-C18	/	96.33	96.41	95.65	95.39	95.43	95.40	96.23	95.36	95.60	95.66
h/H	/	0.64	0.74	0.76	0.81	0.80	0.76	0.69	0.69	0.62	0.66

Note: ^$^ represents the C ratio, referring to the number of carbon atoms (C) and double bonds (N). CTRL = control, SW = scotta wastewater, BMW = buttermilk wastewater, DWW = dairy wastewater, SFAs = saturated fatty acids, UFAs = unsaturated fatty acids, MUFAs = monounsaturated fatty acids, PUFAs = polyunsaturated fatty acids, h/H = hypocholesterolemic/hypercholesterolemic ratio. Mean differences were analyzed using two-way ANOVA with Tukey’s multiple comparisons test. Means denoted by same letter did not differ significantly at *p* ≤ 0.05, while different letters denote, for statistical differences, at least 95% confidence according to Tukey’s multiple comparisons test.

**Table 3 life-15-00184-t003:** Effect of mixotrophy on Thrombogenicity Index (TI), Atherogenicity Index (AI), and hypocholesterolemic/hypercholesterolemic (h/H) ratio by various microalgae strains.

Microalgae	Organic Source	TI	AI	h/H Ratio	Reference
*Limnospira platensis*	dairy	0.86	0.97	0.81	This work
*Limnospira platensis*	photoautotrophy	1.60	0.70	0.60	[51]
*Limnospira platensis*	photoautotrophy	1.46	1.1	0.66	[52]
*Limnospira platensis*	dairy	0.94	0.84	1.07	[23]
*Limnospira platensis*	brewery	3.51	1.76	0.74	[27]
*Parachlorella kessleri* (formerly *Chlorella kessleri*)	glucose	1.51	1.64	1.47	[53]
*Chlorella vulgaris*	molasses	0.79	0.71	2.67	[54]
*Chlorella vulgaris*	glucose	0.42	0.40	2.80	[53]
*Chlorella vulgaris*	glucose	0.38	0.39	2.36	[55]
*Chlorella vulgaris*	brewery	0.48	1.21	2.55	[24]
*Chlorella vulgaris*	dairy	0.59	1.77	1.86	[27]
*Chlorella sorokiniana*	glucose	0.31	0.45	1.76	[55]
*Chlorella sorokiniana*	glucose	0.42	0.49	2.00	[55]
*Chromocloris zofingensis*	molasses	0.40	0.23	3.73	[22]
*Chromocloris zofingensis*	dairy	0.40	0.21	4.22	[22]
*Nannochloropsis oceanica*	photoautotrophy	0.30	0.60	1.44	[51]
*Tetradesmus dimorphus* (formerly *Scenedesmus dimorphus*)	glucose	4.00	1.68	1.07	[56]
*Tetradesmus obliquus* (formerly *Scenedesmus obliquus*)	sodium acetate	-	-	2.09	[57]
*Tetraselmis chui*	photoautotrophy	0.20	0.40	1.04	[51]
*Tribonema aequale*	glucose	0.18	1.02	3.70	[58]

## Data Availability

Data will be available upon request.

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
