# Peer review of "Dairy Wastewaters to Promote Mixotrophic Metabolism in *Limnospira* (*Spirulina*) *platensis*: Effect on Biomass Composition, Phycocyanin Content, and Fatty Acid Methyl Ester Profile"

_life, 2025, doi:10.3390/life15020184_

Round 1

Reviewer 1 Report

Comments and Suggestions for Authors

1. The progress of research on the utilization of agricultural by-products for microalgae cultivation in the introduction is not detailed enough. The following references can help enrich the research background. doi.org/10.3390/foods13213439; 10.1515/chem-2019-0006.

2. Line 136 can be omitted as it is not important information related to the paper.

3. How was the concentration of cheese whey determined? Why were 1%, 2%, and 4% selected?

4. Table 2 is more suitable to be submitted as supplementary information rather than as a separate table in the main text.

5. What is the fitting coefficient of the regression equation in Line 173? What is the applicable OD680 range?

6. Line 199, calcium dichloride???

7. The description of the method in Section 2.6 is overly detailed and needs to be simplified and condensed.

8. Section 3.1 is too verbose. The content of the first few paragraphs is repetitive with the introduction. Focus on the results of your research instead.

9. It would be more reasonable to present Figure 1 as a growth curve. Additionally, since you have obtained the regression curve corresponding to OD and dry weight, why not convert it to dry weight? Also, the spectrophotometer count becomes nonlinear when it exceeds 1, but the OD680 values shown here have reached 3.5.

10. For Figure 2, I suggest changing the statistical method and significance markers to inter-group comparisons rather than comparisons between the experimental group and the control group, as the differences between different concentrations of CW are also of interest to us. Therefore, the statistical method should be adjusted, and the significance markers should be changed to the form of “abcd ”(different letters indicate significant differences) instead of significance asterisk (compared to the control group). Similarly, all significance analyses need to be redone.

11. The same applies to Table 3.

12. The data in the entire paper is not abundant, but the length is too long. The paper lacks conciseness, and there is a lot of redundancy in many places. It is recommended to condense the entire paper.

Author Response

Reviewer 1

  1. The progress of research on the utilization of agricultural by-products for microalgae cultivation in the introduction is not detailed enough. The following references can help enrich the research background. doi.org/10.3390/foods13213439; 10.1515/chem-2019-0006.

Answer: We thank the Reviewer for his advice which is welcomed. Accordingly, we introduced in the Introduction a brief section dealing with the recent interest in utilizing agro and food wastes as supplement for microalgae culture media preparation. In this regard, the two references suggested by the Reviewer have been implemented.

  1. Line 136 can be omitted as it is not important information related to the paper.

Answer: We thank the reviewer for his/her suggestion. Accordingly the content of this phrase related to the reference for the culture medium has been removed.

  1. How was the concentration of cheese whey determined? Why were 1%, 2%, and 4% selected?

Answer: We appreciate the Reviewer thoughtful comments regarding the choice to selected  the concentrations of dairy effluents (DWW) used in our experiments. Our decision to use these dilutions was carefully considered, taking into account several critical factors, including the high organic content of the effluent (Tabel 1) and the associated risks of bacterial contamination. Previous studies, such as the one conducted by Papadopoulos et al. (2022), successfully used higher concentrations of brewery wastewater (BWW), even untreated, for the cultivation of A. platensis. However, these studies typically dealt with specific conditions or strains that may have had a different tolerance to the organic load and potential contaminants present in BWW. In our opinion, high concentration of organic matter in DWW (as in BWW) can lead to rapid bacterial proliferation, even after autoclaving, due to residual bacterial load in the non-axenic inoculum. A. platensis cultures, particularly those not axenic, are vulnerable to competition from bacteria that thrive in nutrient-rich environments. This can negatively impact the growth and biomass productivity of the target microalga. While autoclaving DWW reduces bacterial contamination, it does not entirely eliminate it, especially when non-axenic inocula are used. The combination of residual bacteria from the inoculum and the high nutrient content of the DWW could lead to contamination that compromises the culture integrity. This risk is exacerbated at higher concentrations of DWW. While we started with a lower % of DWW to mitigate these risks, we acknowledge the importance of exploring higher concentrations. However, our preliminary trials suggested that the balance between maintaining culture purity and optimizing growth is delicate, especially with non-axenic strains. Therefore, while we have referenced relevant literature, our approach has been cautious, aiming to establish a baseline that ensures culture stability before testing higher concentrations.

Papadopoulos, KP, Economou CN, Markou G, Nicodemou A, Koutinas M, Tekerlekopoulou AG, Vayenas DV. Cultivation of Arthrospira platensis in Brewery Wastewater. Water 2022, doi:10.3390/w14101547.

  1. Table 2 is more suitable to be submitted as supplementary information rather than as a separate table in the main text.

Answer: We thank the reviewer for his/her suggestion. Accordingly, we moved this Table in the Supplementary Material as S1 Table.

  1. What is the fitting coefficient of the regression equation in Line 173? What is the applicable OD680 range?

Answer: We thank the Reviewer for his/her question which allow us to provide this information. The fitting coefficient of the regression equation is 1.037, as shown in the equation y = 1.037x. This equation describes the relationship between biomass content (g/L) and OD680 (Absorbance at 680 nm), with an R2 value of 0.9523, indicating a strong linear correlation. The applicable OD680 range for this regression is between 0.0 and 2.7, as these were the values obtained during the experimental measurements. It is important to note that when the OD680 readings exceeded 1.0, a 1:1 dilution of the sample was performed to ensure accurate readings within the linear range of the spectrophotometer. After the dilution, the recorded OD value was adjusted by multiplying it by the dilution factor (e.g., by a factor of 2 for a 1:1 dilution) to accurately reflect the original sample's absorbance. This approach ensured that all measurements remained within the spectrophotometer's reliable range while maintaining consistency in the biomass calculations.

  1. Line 199, calcium dichloride???

Answer: We thank the reviewer for letting us know this mistake. Actually the right word for this chemical is calcium chloride instead of calcium dichloride. Accordingly, we corrected it in the revised version of the manuscript.

  1. The description of the method in Section 2.6 is overly detailed and needs to be simplified and condensed.

Answer: We welcome the reviewer’s request. Accordingly, this entire section has been condensed in the revised version of the manuscript. We provided also a reference which can help the readers to find a detailed description of the whole protocol adopted.

  1. Section 3.1 is too verbose. The content of the first few paragraphs is repetitive with the introduction. Focus on the results of your research instead.

Answer: We welcome the reviewer’s comment. Taking it into account, we moved the first part of the Section 3.1 at the very beginning of the Discussion. This contribute also to refine the information on cheese effluents composition helping to detail the discussion on mixotrophic cultivation of cyanobacteria.

  1. It would be more reasonable to present Figure 1 as a growth curve. Additionally, since you have obtained the regression curve corresponding to OD and dry weight, why not convert it to dry weight? Also, the spectrophotometer count becomes nonlinear when it exceeds 1, but the OD680 values shown here have reached 3.5.

Answer: We welcome the reviewer’s suggestion. Taking it into account, we changed this figure with a growth curve showing DW (g/L) obtained by converting OD.

  1. For Figure 2, I suggest changing the statistical method and significance markers to inter-group comparisons rather than comparisons between the experimental group and the control group, as the differences between different concentrations of CW are also of interest to us. Therefore, the statistical method should be adjusted, and the significance markers should be changed to the form of “abcd ”(different letters indicate significant differences) instead of significance asterisk (compared to the control group). Similarly, all significance analyses need to be redone.

Answer: We thank the Reviewer for his/her suggestion. Accordingly, we changed the statistical method and significance markers with letters instead of asterisks. We proceed also to a new analysis of all significances. The same apply also for Figure 3 and 4.

  1. The same applies to Table 3.

Answer: We thank the Reviewer for his/her suggestion. Accordingly, we changed the statistical method and significance markers with letters instead of asterisks. We proceed also to a new analysis of all significances. The same apply also for Figure 3 and 4.

  1. The data in the entire paper is not abundant, but the length is too long. The paper lacks conciseness, and there is a lot of redundancy in many places. It is recommended to condense the entire paper.

Answer: We thank the Reviewer for his/her comment. Accordingly, the whole paper has been revised with the aim to have it shortened and condensed

Reviewer 2 Report

Comments and Suggestions for Authors

Dear Authors,

The authors have a wealth of interesting data, but there is still a key point I do not understand. They repeatedly state that the cultivation conditions are mixotrophic, but the presence of carbon does not necessarily mean that the microalga is assimilating it. They need to provide evidence that spirulina is utilizing a carbon source from these lactic residues. Until this issue is resolved, I do not believe the article makes sense.

Majors:

-I find the title somewhat long; I would advise the authors to shorten it

-L565: “Mixotrophic cultures”??? When the authors mention mixotrophic conditions, which carbon source is Spirulina supposed to be assimilating? Is it the lactose present in dairy waste? That cannot be, as Spirulina does not assimilate lactose. So, what is it? This is a key point that the authors need to clarify-

-How do they know that the conditions are truly mixotrophic? What carbon source is being assimilated? Have they quantified it? Is there acetate in the dairy waste? Is there glycerol? Have they quantified it?

-Why have the authors not analyzed the composition of the different types of dairy residues before and after the growth of spirulina? This would have provided fundamental data to determine which carbon source spirulina is utilizing. How can they know which one it is using if they haven't analyzed it? Please justify. For example, the data in Table 1 are very interesting, but they pertain to before the growth of spirulina. Have the authors also quantified what happens after the growth of spirulina? If not, why?

-Have the authors studied whether light is necessary for the growth of spirulina in these dairy residues? In other words, does spirulina grow in the dark with the dairy resiudues? This would indicate that it is truly growing heterotrophically using a carbon source from the residues. Have they tested this? Include this discussion in the text, please.

-L44: I would also suggest including information about biofuels

-It is not clear to me that Spirulina can be considered a microalga; at the very least, this is a topic open to debate. It should be mentioned that it is a photosynthetic prokaryotic organism. In the introduction, it would be helpful to state that there are other microalgae, such as Chlorella or Chlamydomonas, on which similar research has also been conducted.

-L138: "10 ml of microalgae" is somewhat imprecise. What concentration? How many cells?

-According to Table 2, why were different amounts of inoculum (60, 40, 24) used depending on the type of CW? Please provide justification.

-L298-L305: Here, as an example, and in other parts of the results section, there is excessive digression. This is the results section, and such content would fit better in the introduction. Please remove anything that is not presenting experimental results, eliminate superfluous information, or move it to the introduction and results section as appropriate.

-L325........: I’m sorry, but this is just another example. The authors are mixing discussion with results. These paragraphs should be removed and moved to the discussion section. As mentioned, the results section requires a thorough revision.

-Figure 1: I don’t think this type of graph is suitable for presenting these results. The statistical error is not shown, and the colors of the symbols are difficult to differentiate. I recommend converting it into a line graph and including the statistical error.

-Figures 3B and 4B do not have statistical error.

Minors:

-L317: “effluents” Typo

Author Response

Reviewer 2

Dear Authors,

The authors have a wealth of interesting data, but there is still a key point I do not understand. They repeatedly state that the cultivation conditions are mixotrophic, but the presence of carbon does not necessarily mean that the microalga is assimilating it. They need to provide evidence that spirulina is utilizing a carbon source from these lactic residues. Until this issue is resolved, I do not believe the article makes sense.

Majors:

-I find the title somewhat long; I would advise the authors to shorten it

Answer: We welcomed the suggestion of the Reviewer. Accordingly we condensed in the title the three types of dairy wastes as dairy wastewaters in general

-L565: “Mixotrophic cultures”??? When the authors mention mixotrophic conditions, which carbon source is Spirulina supposed to be assimilating? Is it the lactose present in dairy waste? That cannot be, as Spirulina does not assimilate lactose. So, what is it? This is a key point that the authors need to clarify-

Answer: We thank the Reviewer for his/her comments that allow us to clarify this point. Taking into account the Reviewer questions we introduced at the very beginning of the Discussion section a detailed description of sugars composition for dairy wastes in general and for the type of effluents used in this work, providing evidence that cheese whey has been previously used by other researchers for the cultivation of A. platensis

-How do they know that the conditions are truly mixotrophic? What carbon source is being assimilated? Have they quantified it? Is there acetate in the dairy waste? Is there glycerol? Have they quantified it?

Answer: We appreciate the Reviewer insightful questions regarding the carbon sources potentially assimilated by Spirulina in the study. This allow us to provide our clarifications to this matter. In our work, Spirulina was cultivated in cheese whey, buttermilk, and dairy wastewater, which are complex substrates containing a mixture of organic and inorganic components. While we did not directly quantify specific organic carbon compounds (such as glycerol or acetate), the observed growth pattern, which exceeds that expected for purely autotrophic conditions, strongly suggests the involvement of mixotrophic metabolism. Previous studies (Narayan et al. 2005, de Morais et al. 2019, Pereira et al. 2019, Shayesteh et al. 2023, have demonstrated that Spirulina can grow mixotrophically using carbon sources such as glycerol, acetate, or sugars other than lactose, which Spirulina cannot metabolize due to the absence of the necessary β-galactosidase enzyme. While measuring the specific organic carbon compounds (such as acetate or glycerol) in the dairy effluents would have provided more direct evidence, this was not performed in the present study since the primary focus of it was to evaluate the feasibility of using dairy effluents as a growth medium for Spirulina, rather than characterizing the effluents in detail. Dairy wastewater composition is highly variable and influenced by the source and processing methods, making it challenging to standardize and measure specific components without extensive preliminary characterization, which was beyond the scope of this study. The growth performance and biomass yield were considered as indirect but reliable indicators of the ability of Spirulina to assimilate organic carbon sources, consistent with mixotrophic metabolism. Dairy waste streams, including cheese whey and buttermilk, have been reported (including the literature cited below) to contain residual organic compounds such as glycerol, acetate, and other small organic acids, which can be utilized by Spirulina in mixotrophic mode. While we did not quantify specific carbon sources like glycerol or acetate in this study, our findings align with existing literature showing that Spirulina can grow mixotrophically in similar effluents containing these compounds. Future work will aim to include detailed compositional analysis of the effluents to directly measure the concentrations of potential organic carbon sources and provide a more comprehensive understanding of the mixotrophic growth mechanisms.

Narayan MS, Manoj GP, Vatchravelu K, Bhagyalakshmi N, Mahadevaswamy M. Utilization of glycerol as carbon source on the growth, pigment and lipid production in Spirulina platensis. Int J Food Sci Nutr. 2005, 521-528. doi: 10.1080/09637480500410085

de Morais, E.G., Druzian, J.I., Larroza Nunes, I., Greque de Morais, M., Vieira Costa, J.A. Glycerol increases growth, protein production and alters the fatty acids profile of Spirulina (Arthrospira) sp LEB 18. Process Biochem 2019, 76, 40-45. https://doi.org/10.1016/j.procbio.2018.09.024.

Pereira MIB, Chagas BME, Sassi R, Medeiros GF, Aguiar EM, Borba LHF, Silva EPE, Neto JCA, Rangel AHN. Mixotrophic cultivation of Spirulina platensis in dairy wastewater: Effects on the production of biomass, biochemical composition and antioxidant capacity. PLoS One 2019, 14(10), e0224294. doi: 10.1371/journal.pone.0224294.

Shayesteh H, Laird DW, Hughes LJ, Nematollahi MA, Kakhki AM, Moheimani NR. Co-Producing Phycocyanin and Bioplastic in Arthrospira platensis Using Carbon-Rich Wastewater. BioTech 2023, 12(3), 49. doi: 10.3390/biotech12030049.

- Why have the authors not analyzed the composition of the different types of dairy residues before and after the growth of Spirulina? This would have provided fundamental data to determine which carbon source Spirulina is utilizing. How can they know which one it is using if they haven't analyzed it? Please justify. For example, the data in Table 1 are very interesting, but they pertain to before the growth of Spirulina. Have the authors also quantified what happens after the growth of Spirulina? If not, why?

Answer: We thank the Reviewer for his/her comments which allow us to clarify our point of view at this regard. We acknowledge the importance of analyzing the composition of dairy residues both before and after the cultivation of Spirulina to fully understand the carbon sources utilized during growth. While such analyses were not performed in this study, our approach focused on demonstrating the feasibility of Spirulina cultivation using various dairy residues as nutrient sources. The primary objective was to assess biomass production, nutrient assimilation efficiency, and overall growth performance under different waste-based media, rather than to conduct a detailed carbon flux analysis. Actually, this research aimed to evaluate Spirulina’s potential for biomass production in cost-effective, waste-based media. The primary focus was on growth metrics (e.g., biomass yield, protein content, lipid accumulation) as indicators of nutrient assimilation, rather than conducting an exhaustive compositional analysis of the medium. In addition, the sugar composition of dairy residues can be highly variable and complex, comprising not only lactose but also minor sugars (e.g., glucose, galactose) and organic acids (e.g. acetate, citrate). The equipment and methods required for such high-resolution sugar profiling (such as HPLC or GC-MS) were beyond the analytical scope and resources of this study. In our opinion, the observed growth of Spirulina in dairy residues strongly suggests nutrient assimilation, with possibly organic acids (e.g. acetate) being utilized as carbon sources. While the exact proportion of each carbon source consumed was not quantified, the significant increase in biomass implies effective utilization of available nutrients. We agree indeed with the Reviewer that a comparative analysis of the medium composition before and after cultivation would provide valuable insights into the metabolic pathways of Spirulina when grown on dairy residues. Therefore, by taking into account it, future studies will aim to incorporate sugar-specific analyses and quantify their depletion during growth to identify the primary carbon sources utilized. Such data would enhance our understanding of the metabolic potential of Spirulina and optimize its application in bioremediation and biomass production.

- Have the authors studied whether light is necessary for the growth of Spirulina in these dairy residues? In other words, does Spirulina grow in the dark with the dairy residues? This would indicate that it is truly growing heterotrophically using a carbon source from the residues. Have they tested this? Include this discussion in the text, please.

Answer: We thank the Reviewer for his/her comments which allow us to clarify our point of view at this regard. We appreciate the insightful suggestion to investigate the necessity of light for Spirulina growth in dairy residues, which would clarify whether the growth observed is truly heterotrophic or mixotrophic. In this study, the focus was on assessing Spirulina’s growth performance and biomass production in dairy waste-based media under conditions that simulate a typical mixotrophic cultivation setup (i.e. light availability combined with an organic carbon source). For several reasons the effect of light was not specifically tested. First, the aim of this study was to evaluate the feasibility of using dairy residues, such as cheese whey and buttermilk, as nutrient-rich media for Spirulina cultivation. The primary focus was on assessing growth performance under light conditions representative of mixotrophic cultivation, which is commonly employed for Spirulina. Investigating heterotrophic growth in the absence of light was outside the scope of this work. In addition, previous studies have demonstrated that Spirulina grows optimally under mixotrophic conditions, where both light and organic carbon sources contribute to its metabolism. Dairy residues, rich in sugars and other organic compounds, provide a potential carbon source, while light supports photosynthetic activity. The significant biomass production observed in this study suggests successful nutrient assimilation in a mixotrophic mode. Although Spirulina is primarily photoautotrophic, some studies indicate its ability to grow under heterotrophic conditions with certain carbon sources (e.g. glucose, glycerol). Exploring the capacity of utilizing one type of sugar or another would require dedicated experiments, including dark-incubation trials and carbon source utilization analyses, which were beyond the current study's objectives and resources. To this aim, by acknowledging the importance of distinguishing between mixotrophic and heterotrophic growth modes to better understand Spirulina’s metabolic pathways, our future studies will be tailored to investigate the role of light explicitly by comparing growth under light and dark conditions using dairy residues.

-L44: I would also suggest including information about biofuels

Answer: According to Reviewer’s request biofuels has been added as another of microalgae filed of application, corroborated by a suitable reference

-It is not clear to me that Spirulina can be considered a microalga; at the very least, this is a topic open to debate. It should be mentioned that it is a photosynthetic prokaryotic organism. In the introduction, it would be helpful to state that there are other microalgae, such as Chlorella or Chlamydomonas, on which similar research has also been conducted.

Answer: We thank the Reviewer for his/her suggestion which allows us to implement his/her requests in the revised version of the manuscript

-L138: "10 ml of microalgae" is somewhat imprecise. What concentration? How many cells?

Answer: The missing information has been provided.

-According to Table 2, why were different amounts of inoculum (60, 40, 24) used depending on the type of CW? Please provide justification.

Answer: We thank the Reviewer for his/her comments which allow us to clarify our point of view at this regard. The different amounts of inoculum (60, 40, 24 mL) used depending on the type of CW and its concentration can be justified by the potential variability in the composition and nutrient profiles of the different CW types. In the selection step of these amounts we consider some aspects about variation in nutritional content across the three types of dairy wastes used and balancing carbon source availability. As for the nutritional content, different CW types used (scotta whey, buttermilk, and final CW wastewater) likely differ in their organic matter, sugar content, or inhibitory compounds (such as high salt or fat concentrations). Adjusting the inoculum volume ensures that the initial cell density is optimized for growth, avoiding excessive stress or lag phase due to nutrient or environmental imbalances. In particular, higher concentrations of CW in the medium (e.g. 4%) may have inhibitory effects due to excessive nutrients, osmotic stress, or the presence of toxic compounds. Lower inoculum volumes for certain CW types (buttermilk wastewater and final CW wastewater) could reduce the initial nutrient competition or stress on the microalgae. As for the carbon availability, scotta whey appears to allow for higher inoculum volumes (60 mL), suggesting it may have a more balanced nutrient profile conducive to higher starting densities. By contrast, buttermilk and final CW wastewater may require lower initial densities (40 or 24 mL) to prevent overloading the culture medium. All this considered, despite varying inoculum volumes, the total culture volume was maintained at 600 mL for all treatments. This ensures consistency in experimental conditions and allows for the direct comparison of results across treatments.

-L298-L305: Here, as an example, and in other parts of the results section, there is excessive digression. This is the results section, and such content would fit better in the introduction. Please remove anything that is not presenting experimental results, eliminate superfluous information, or move it to the introduction and results section as appropriate.

Answer: We welcome the reviewer’s comment. Taking it into account, we moved this part into the Discussion section helping to refine the information on cheese effluents composition.

-L325........: I’m sorry, but this is just another example. The authors are mixing discussion with results. These paragraphs should be removed and moved to the discussion section. As mentioned, the results section requires a thorough revision.

Answer: We welcome the reviewer’s comment. Accordingly, we moved this part into the Discussion section.

-Figure 1: I don’t think this type of graph is suitable for presenting these results. The statistical error is not shown, and the colors of the symbols are difficult to differentiate. I recommend converting it into a line graph and including the statistical error.

Answer: We thank the Reviewer for his/her comment about the suitability of this graph. According to also a comment at this regard by the Reviewer # 1, we decided to substitute the growth curve expressed by OD in a new graph where the curves depict the dry weight, which has been obtained by converting OD values, as explained previously

-Figures 3B and 4B do not have statistical error.

Answer: We thank the Reviewer for letting us know this inaccuracy. In the revised version of the manuscript these mistakes have been corrected accordingly

Minors:

-L317: “effluents” Typo

Answer: This mistake has been corrected

Reviewer 3 Report

Comments and Suggestions for Authors

The manuscript investigates very well the application of dairy byproducts as additives for the microalgae growth, in order to define a process economically sustainable. These findings are indeed helpful for the application of microalgae in this field, however I would clarify the following points:

1.        Line 152: In the caption of Table 1 there is a mistake, it is written chemical organic demand and not oxygen. Please modify.

2.        Line 148: it would be worth it to have also the composition of the cheese whey samples before filtration and sterilization, so that to understand if these procedures affected the composition of them, and moreover, to better understand the feasibility of the utilization of microalgae in a real case scenario, since both processes are quite expensive and difficult to apply on industrial scale levels.

3.        Can you please point out which was the room temperature of your experiments? It might be interesting to evaluate the feasibility of the process with higher temperatures, since the dairy wastewater should have high temperatures due to the steps in which is involved.

4.        Did you think about performing, as additional control experiment, mixotrophic grow for Spirulina by adding the main carbon source present in the same %? It would be helpful to understand which is the benchmark for Spirulina in mixotrophic growth conditions.

5.        Line 517, 520, 522: Please modify to Italic for A. Platensis.

6.        I am missing in the discussion an overall evaluation on the feasibility of the process, in terms of scalability and economic viability. Unfortunately, under laboratory conditions, microalgae are better performers.

Author Response

Reviewer 3

The manuscript investigates very well the application of dairy byproducts as additives for the microalgae growth, in order to define a process economically sustainable. These findings are indeed helpful for the application of microalgae in this field, however I would clarify the following points:

  1. Line 152: In the caption of Table 1 there is a mistake, it is written chemical organic demand and not oxygen. Please modify.

Answer: We thank the Reviewer for letting us know this mistake. Accordingly, it has been corrected in the revised version of the manuscript

  1. Line 148: it would be worth it to have also the composition of the cheese whey samples before filtration and sterilization, so that to understand if these procedures affected the composition of them, and moreover, to better understand the feasibility of the utilization of microalgae in a real case scenario, since both processes are quite expensive and difficult to apply on industrial scale levels.

Answer: We appreciate the Reviewer's insightful suggestion regarding the analysis of cheese whey composition. Unfortunately, this analysis was not included in the scope of the present study due to logistical and resource constraints. However, we acknowledge the importance of understanding how these procedures might influence the composition of the cheese whey and their potential implications for the utilization of microalgae in large scale applications. While filtration and sterilization are widely used to standardize substrate quality and eliminate contaminants in laboratory studies, we agree that these processes can be challenging to implement at an industrial scale due to cost and complexity. In future studies, we plan to explore alternative approaches that minimize preprocessing or incorporate raw cheese whey to better evaluate the feasibility of industrial-scale applications. We recognize indeed that the potential impact of filtration and sterilization on the composition of cheese whey and, consequently, on microalgal growth could provide valuable insights. To this end, future research from our side will take into account also the chemical and nutritional changes induced by these procedures by assessing their effect on microalgae cultivation systems.

  1. Can you please point out which was the room temperature of your experiments? It might be interesting to evaluate the feasibility of the process with higher temperatures, since the dairy wastewater should have high temperatures due to the steps in which is involved.

Answer: We thank the Reviewer for their observation regarding the temperature conditions of the experiments. The cultivation experiments were conducted at room temperature during the fall season (October-November) in a controlled cultivation room. The temperature within the room was maintained at a stable level due to the warmth provided by the LED lighting system installed around ten 500 L photobioreactors. Although the exact room temperature was not recorded during the experiments, we estimate that it was approximately 26 ± 1°C based on typical environmental conditions and the heating effect of the LEDs. We agree with the Reviewer that evaluating the feasibility of the process at higher temperatures could provide valuable insights, especially considering that dairy wastewater is typically discharged at elevated temperatures during industrial processes. Future studies could explore the impact of higher cultivation temperatures on microalgal growth and system efficiency, simulating conditions closer to those encountered in industrial scenarios. This would help to assess the process's adaptability and efficiency under conditions more representative of real-world applications.

  1. Did you think about performing, as additional control experiment, mixotrophic grow for Spirulina by adding the main carbon source present in the same %? It would be helpful to understand which is the benchmark for Spirulina in mixotrophic growth conditions.

Answer: We thank the reviewer for the suggestion of performing additional control experiments to test mixotrophic growth by adding the main carbon source present in the same percentages as in the dairy wastewater (CW). While this approach was not included in the current study, it is indeed an excellent idea for benchmarking the performance of Spirulina under controlled mixotrophic conditions. In the present setup, as shown in Table S1, various percentages of CW were amended to the culture medium (JM) to evaluate Spirulina's growth under different concentrations of dairy wastewater. However, we acknowledge that a direct comparison with mixotrophic conditions using isolated carbon sources (e.g. glucose, which is between the primary components of CW) would provide valuable insights into the specific contributions of organic carbon to Spirulina growth. Future experiments could focus on systematically assessing mixotrophic growth by adding the main carbon source(s) of CW at equivalent concentrations (e.g., 1%, 2%, and 4%) to directly compare the growth rates and productivity with those observed in wastewater-based cultures. Such experiments would also help elucidate whether additional nutrients or interactions within CW enhance Spirulina growth beyond the effect of the carbon source alone.

  1. Line 517, 520, 522: Please modify to Italic for platensis.

Answer: We thank the Reviewer for letting us know these inaccuracies. Accordingly, they have been corrected in the revised version of the manuscript

  1. I am missing in the discussion an overall evaluation on the feasibility of the process, in terms of scalability and economic viability. Unfortunately, under laboratory conditions, microalgae are better performers.

Answer: We thank the Reviewer for his/her comment. Accordingly, we added at the end of Discussion a brief section pertinent to feasibility of the whole process.

Round 2

Reviewer 1 Report

Comments and Suggestions for Authors

All concerns have been addressed properly. It meets the standard of publication.

Author Response

We thank the Editor for his/her request to revise the while manuscript addressing minor revisions on taxonomy and nomenclature. Therefore, in the revised version of the manuscript all the scientific names of the cited strains have been corrected (in red) accordingly.

Reviewer 2 Report

Comments and Suggestions for Authors

I believe the authors have adequately addressed all of my comments and suggestions.

Author Response

(The authors gave the same response as above.)
